# DynaIP: Dynamic Image Prompt Adapter for Scalable Zero-shot Personalized Text-to-Image Generation

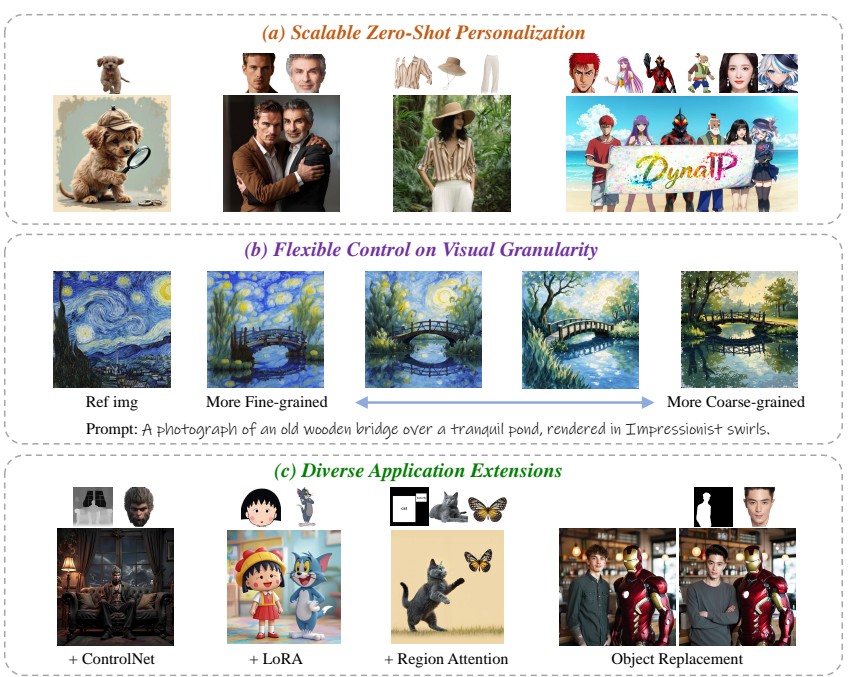

Figure 1: Representative results showcase the capabilities of DynaIP in: (a) **Scalable zero-shot personalized text-to-image generation**—spanning single-subject to multi-subject—*trained solely on single-subject datasets*. (b) **Flexible control on the visual granularity of concept preservation**, enabled by modulating fusion coefficients for image features across hierarchical levels. (c) **Native compatibility with base model extensions**, unlocking diverse application scenarios.

## Abstract

Personalized Text-to-Image (PT2I) generation aims to produce customized images based on reference images. A prominent interest pertains to the integration of an image prompt adapter to facilitate zero-shot PT2I without test-time fine-tuning. However, current methods grapple with three fundamental challenges: **1.** the elusive equilibrium between Concept Preservation (CP) and Prompt Following (PF), **2.** the difficulty in retaining fine-grained concept details in reference images, and **3.** the restricted scalability to extend to multi-subject personalization. To tackle these challenges, we present ***Dynamic Image Prompt Adapter (DynaIP)***, a cutting-edge plugin to enhance the *fine-grained concept fidelity*, *CP·PF balance*, and *subject scalability* of state-of-the-art T2I multimodal diffusion transformers (MM-DiT) for PT2I generation. *Our key finding is that MM-DiT inherently exhibit decoupling learning behavior when injecting reference image features into its dual branches via cross attentions.* The noisy image branch selectively captures the concept-specific information of the reference image, while the text branch learns concept-agnostic information. Based on this, we design an innovative *Dynamic Decoupling Strategy* that removes the interference of concept-agnostic information during inference, significantly enhancing the CP·PF balance and further

bolstering the scalability of multi-subject compositions. Moreover, we identify the visual encoder as a key factor affecting fine-grained CP and reveal that *the hierarchical features of commonly used CLIP can capture visual information at diverse granularity levels.* Therefore, we introduce a novel *Hierarchical Mixture-of-Experts Feature Fusion Module* to fully leverage the hierarchical features of CLIP, remarkably elevating the fine-grained concept fidelity while also providing flexible control of visual granularity. Extensive experiments across single- and multi-subject PT2I tasks verify that our DynaIP outperforms existing approaches, marking a notable advancement in the field of PT2l generation.

# 1 INTRODUCTION

Recent advances in denoising diffusion models (Ho et al., 2020) have led to remarkable success in Text-to-Image (T2I) generation, where State-of-the-Art (SOTA) models (Esser et al., 2024; Labs, 2024) can produce visually plausible, high-quality images based on textual prompts. Subsequent researches have extended T2I to Personalized Text-to-Image (PT2I) generation (Gal et al., 2023; Ruiz et al., 2023; Shi et al., 2024), which aims to synthesize customized images based on both reference images and text prompts, ensuring that objects (*e.g.*, persons, animals) in the generated images maintain specific concept characteristics.

Pioneering PT2I methods, represented by DreamBooth (Ruiz et al., 2023) and Textual Inversion (Gal et al., 2023), primarily relied on fine-tuning models or specific text embeddings, often suffering from low efficiency. To reduce usage cost, recent attention has shifted towards finetuning-free paradigms, as exemplified by IP-Adapter (Ye et al., 2023) and OminiControl (Tan et al., 2025), which achieve zero-shot personalization via engineered decoupled cross-attention or multimodal joint attention mechanisms. Furthermore, there are several works (Xiao et al., 2024a; Wang et al., 2025; Huang et al., 2025) have extended Single-Subject PT2I (SS-PT2I) to Multi-Subject PT2I (MS-PT2I).

In this work, we are interested in the adapter-based methods (Ye et al., 2023; Wang et al., 2025; Huang et al., 2025; Kong et al., 2025) owing to their intrinsic high flexibility and low computational complexity. Methods within this paradigm generally extract features from reference images via a visual encoder (Radford et al., 2021) and subsequently inject them into the cross-attention layers of a T2I diffusion model. Although substantial advancements have been made, this line of approaches are still confronted with three pivotal limitations:

1. Entanglement of concept-specific information (*e.g.*, ID, shape, and textures) and concept-agnostic information (*e.g.*, posture, perspective, and illumination) in the injected reference image features, leading to an irreconcilable trade-off between Concept Preservation (CP, image & image consistency) and Prompt Following (PF, image & prompt consistency) in generated results (Peng et al., 2025; He et al., 2025), as shown in Fig. 2 (a).

2. Insufficient fine-grained feature extraction from reference images, resulting in failure to faithfully recover the intricate object details in customized outputs, as shown in Fig. 2 (b).

3. Significant challenge to directly extend the capability of SS-PT2I to MS-PT2I. Existing approaches often heavily rely on well-curated large-scale multi-subject datasets (Xiao et al., 2024a; Wang et al., 2025), or encounter pronounced inconsistency when composing multiple subjects via mask-guided feature injection (Ye et al., 2023; Team, 2024), severely impeding their scalability, as shown in Fig. 2 (c).

Facing the aforementioned challenges, in this work, we propose the ***Dyna**mic **I**mage **P**rompt Adapter (DynaIP)*, aiming at improving the *concept fidelity*, *CP·PF balance*, and *subject scalability* of adapter-based methods for PT2I generation tasks. Our DynaIP serves as a cutting-edge plugin for SOTA T2I multimodal diffusion transformers (MM-DiT) (Esser et al., 2024; Labs, 2024) and is compatible with diverse extensions (*e.g.*, ControlNet (Zhang et al., 2023), LoRA (Hu et al.), and Region Attention (Chen et al., 2024a)) of the same base model, as demonstrated in Fig. 1 (c).

The key insight is that *the latest MM-DiT architecture inherently exhibits decoupling learning behavior when injecting reference image features into its dual branches via cross attentions.* Specifically, the noisy image branch selectively captures the concept-specific information of the reference

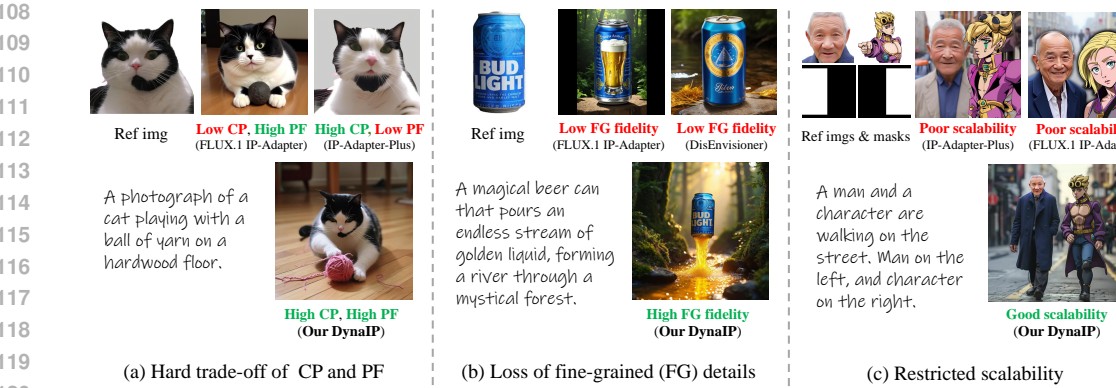

Figure 2: **Limitations of existing adapter-based PT2I methods** (*e.g.*, (Ye et al., 2023; Team, 2024; He et al., 2025)), including (a) irreconcilable trade-off between CP and PF, (b) loss of fine-grained concept details, and (c) restricted scalability to directly extend SS-PT2I to MS-PT2I via mask-guided feature injection. Our proposed DynaIP addresses all these challenges.

image, such as the subject's ID and unique appearance, whereas the text branch learns the concept-agnostic information, such as posture, perspective, and illumination. Based on this finding, we design an innovative *Dynamic Decoupling Strategy (DDS)* that removes the interference of concept-agnostic information during the inference phase. This dynamic isolation offers two key advantages: (1) *improving the prompt following while maintaining the concept preservation, and thus achieving a better sweet spot between them*, and (2) *mitigating the visual integration inconsistency arising from the retention of concept-agnostic information when extending SS-PT2I to MS-PT2I, thereby significantly boosting the subject scalability*.

Moreover, we believe that the key factor affecting fine-grained CP of reference images lies in the visual encoder. Existing methods (Ye et al., 2023; Xiao et al., 2024a; Wang et al., 2025; Huang et al., 2025) typically employ the CLIP (Radford et al., 2021) image encoder to extract relatively high-level information from deep features of the final or penultimate layers. Unfortunately, these deep features incur substantial loss of detailed information, rendering them inadequate for recovering the fine-grained visual details of reference images. To address this gap, we first systematically investigate the reconstruction capabilities of CLIP's multi-layer features and reveal that *these hierarchical features can capture visual information at diverse granularity levels*. Building on this insight, we further propose a novel *Hierarchical Mixture-of-Experts Feature Fusion Module (HMoE-FFM)* to fully harness CLIP's hierarchical features. HMoE-FFM deploys layer-specific expert networks to process hierarchical features and dynamically calibrates their fusion coefficients via a routing mechanism that adapts to the characteristics of each reference image. This approach not only enhances the fidelity of fine-grained concept details but also facilitates precise modulation of visual granularity. For instance, users can manually adjust the fusion coefficients of experts' outputs, thereby *enabling flexible control over the granularity of concept preservation* according to specific needs, as demonstrated in Fig. 1 (b) and Sec. B.4.

Overall, the contributions of our proposed DynaIP can be summarized as follows:

- We identify an intrinsic decoupling learning behavior of the MM-DiT architecture and, building on this insight, devise a *Dynamic Decoupling Strategy* to effectively disentangle concept-specific information from concept-agnostic information within reference images. This strategy significantly enhancing the CP·PF balance and further bolstering the scalability of multi-subject compositions.

- We reveal that the hierarchical features of standard CLIP inherently encode visual information across multiple granularities. Based on this, we propose a novel *Hierarchical Mixture-of-Experts Feature Fusion Module* that dynamically integrates these multi-level features to preserve both fine-grained visual details and semantic consistency. This module not only enhances concept fidelity but also enables flexible control over visual granularity.

- Extensive experiments demonstrate the superior performance and broad application potential of DynaIP. Our method outperforms SOTA approaches in attaining *fine-grained con-*

*cept fidelity* and striking an *optimal balance between CP and PF*. Moreover, it exhibits exceptional *scalability*: being *directly extendable to MS-PT2I scenarios without requiring additional training on multi-subject datasets*, as demonstrated in Fig. 1 (a).

## 2 RELATED WORK

Please refer to Sec. A for related work.

## 3 PRELIMINARY

### 3.1 MULTIMODAL DIFFUSION TRANSFORMER (MM-DiT)

MM-DiT was first proposed in Stable Diffusion 3 (Esser et al., 2024) and later applied in FLUX.1 (Labs, 2024). It enhances the vanilla DiT (Peebles & Xie, 2023; Chen et al., 2024c) by using a unified Multi-Modal Attention (MMA) mechanism to jointly process noisy image tokens $X \in \mathbb{R}^{m \times d}$ and text tokens $T \in \mathbb{R}^{n \times d}$. Here, $d$ represents the latent dimension, while $m$ and $n$ represent the sequence lengths of image and text tokens, respectively.

Specifically, the MMA mechanism projects text and image tokens into position-encoded query $\mathbf{Q} = [\mathbf{Q}_T, \mathbf{Q}_X]$, key $\mathbf{K} = [\mathbf{K}_T, \mathbf{K}_X]$, and value $\mathbf{V} = [\mathbf{V}_T, \mathbf{V}_X]$ representations, enabling cross-modal attention computation across all tokens:

$$MMA([T, X]) = softmax(\frac{\mathbf{Q}\mathbf{K}^\top}{\sqrt{d}})\mathbf{V} = [T^{MMA}, X^{MMA}], \tag{1}$$

where $[\cdot, \cdot]$ denotes the sequence concatenation. $T^{MMA}$ and $X^{MMA}$ are the new text and noisy image tokens after MMA, respectively. The MMA allows both representations to operate within their respective spaces while still taking the other into account.

### 3.2 IMAGE PROMPT ADAPTER FOR MM-DiT

Image Prompt Adapter (IP-Adapter) (Ye et al., 2023) is a lightweight method to endow pretrained T2I diffusion models with the ability to utilize image prompts. Its core lies in a decoupled cross-attention mechanism that distinctly separates the cross-attention layers for processing text features and image features. The majority of prior works have predominantly relied on U-Net-based diffusion models. When transferring to MM-DiT-based models, a vanilla solution (Team, 2024) is computing the Cross-Attention (CA) between noisy image tokens $X$ and reference image tokens $C \in \mathbb{R}^{h \times d}$ (Eq. (2)), as illustrated in Fig. 3 (a-b). Here, $h$ represents the sequence length of reference image tokens, which is typically extracted by a feature extraction model comprising a pretrained image encoder (*e.g.*, CLIP (Radford et al., 2021)) and a trainable projection network.

$$CA(X, C) = softmax(\frac{\mathbf{Q}_X \mathbf{K}_C^\top}{\sqrt{d}})\mathbf{V}_C, \tag{2}$$

where $\mathbf{Q}_X$ is the query representation of noisy image tokens $X$ in Eq. (1), while $\mathbf{K}_C$ and $\mathbf{V}_C$ are newly learned key and value representations obtained by linearly projecting the reference image tokens $C$.

Then, the Decoupled Cross-Attention (DCA) mechanism integrates the output of image CA with the output of text CA:

$$DCA(T, X, C) = X^{MMA} + \lambda \cdot CA(X, C). \tag{3}$$

Here, the output of text CA in MM-DiT is embedded in the new noisy image tokens $X^{MMA}$ after MMA (Eq. (1)). $\lambda$ is a weight factor to control the influence of the reference image features.

## 4 DYNAMIC IMAGE PROMPT ADAPTER

As discussed in Sec. 1, current adapter-based methods face fundamental challenges in fine-grained concept fidelity, CP-PF balance, and subject scalability. To tackle this challenges, we introduce

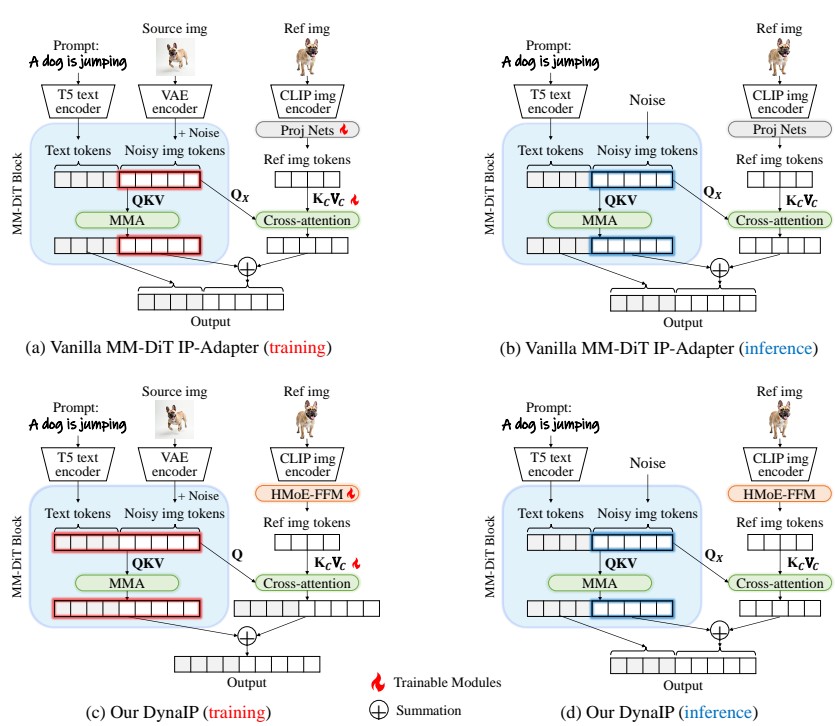

Figure 3: Training and inference pipeline of **(a-b)** vanilla IP-Adapter and **(c-d)** our DynaIP.

*DynaIP*, which comprises two key components. First, to improve CP·PF balance and subject scalability, we propose *Dynamic Decoupling Strategy*. It leverages an intrinsic characteristic of MM-DiT to dynamically disentangle concept-specific information from concept-agnostic information in reference images. Second, we present *Hierarchical Mixture-of-Experts Feature Fusion Module*, which dynamically integrates multi-level CLIP features according to different reference inputs. It not only significantly boosts fine-grained concept fidelity but also enables flexible control over visual granularity. The subsequent sections elaborate on these core components in detail.

## 4.1 DYNAMIC DECOUPLING STRATEGY

As introduced in Sec. 3.2 and Fig. 3 (a-b), the vanilla IP-Adapter for MM-DiT holistically injects reference image features into the noisy image, inevitably introducing concept-agnostic information that undermines personalization quality, as shown in Fig. 2. Considering that the *text branch in MM-DiT interacts with the noisy image branch through MMA (Eq. (1)) to govern the overall semantics, actions, and visual presentation of the generated image—all of which are irrelevant to specific concept information*—our key strategy is to enable reference image features to perform cross-attention with both the noisy image branch and the text branch simultaneously in the training phase, as illustrated in Fig. 3 (c) and formalized below:

$$CA([T, X], C) = softmax(\frac{\mathbf{Q}\mathbf{K}_C^\top}{\sqrt{d}})\mathbf{V}_C. \tag{4}$$

Subsequently, the output of CA is added to the output of MMA (Eq. (1)):

$$DCA'(T, X, C) = [T^{MMA}, X^{MMA}] + \lambda \cdot CA([T, X], C). \tag{5}$$

In this way, the noisy image branch focuses on capturing the concept-specific information of the reference image, such as the subject's ID and unique appearance, while the text branch specializes in learning the concept-agnostic information like posture, perspective, and illumination (see detailed analyses in Sec. B.6). Thus, during inference, we can dynamically remove concept-agnostic information by performing CA exclusively with the noisy image branch, as in Fig. 3 (d) and Eqs. (2, 3). This simple yet effective strategy significantly improves the CP·PF balance and bolsters the scalability of multi-subject compositions, as will be demonstrated in later Sec. 5.4.

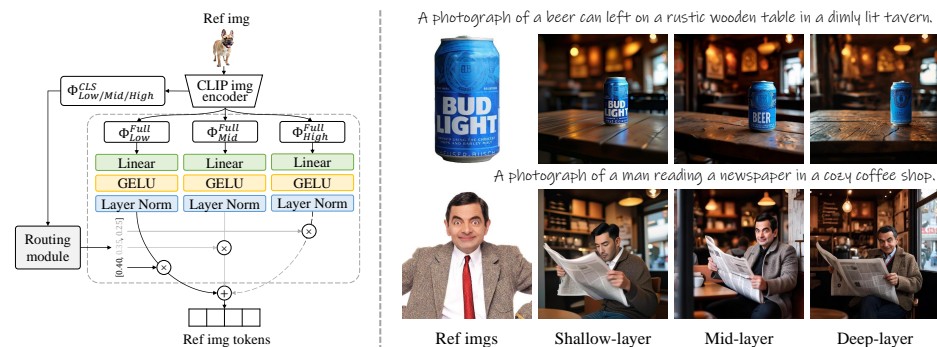

Figure 4: **Left**: Architecture of our proposed HMoE-FFM. **Right**: Personalization results generated by injecting features from different layers of CLIP via cross-attentions, demonstrating that CLIP's hierarchical features can capture visual information at diverse granularity levels.

## 4.2 HIERARCHICAL MIXTURE-OF-EXPERTS FEATURE FUSION

The image encoder used in IP-Adapter plays a critical role to extract concept information from reference images. Current methods typically utilize CLIP (Radford et al., 2021), extracting features from its final or penultimate layers. While a few works (Team, 2024; Kong et al., 2025) have explored more powerful alternatives such as SigLIP (Zhai et al., 2023) and DINOv2 (Oquab et al., 2024), they still rely on deep-layer features. These deep-layer features suffer from substantial loss of detailed information, making them inadequate for recovering fine-grained visual details of reference images. To address this limitation, we first investigate CLIP's feature extraction capabilities by systematically visualizing the reconstruction performance of its multi-layer features—achieved by injecting each layer's features individually via cross-attentions. As demonstrated in Fig. 4-right, we made a striking observation: *CLIP's hierarchical features excel at capturing visual information across varying granularity levels*. Specifically, shallow-layer features (e.g., layer 10) capture low-level patterns such as lines and text; mid-layer features (e.g., layer 17) capture fine-grained textures and structures like facial details; and deep-layer features (e.g., layer 24) capture semantic information alongside coarse-grained textures and structures (see results across more layers in Sec. B.8). Unfortunately, these multi-granularity hierarchical features have not been fully utilized in existing works.

To bridge this gap and fully harness the multi-granularity potential of CLIP's hierarchical features, we further propose a novel *Hierarchical Mixture-of-Experts Feature Fusion Module (HMoE-FFM)*, as depicted in Fig. 4-left. Specifically, let $\Phi_{Low}$, $\Phi_{Mid}$, and $\Phi_{High}$ denote the image features from CLIP's shallow, middle, and deep layers, respectively. $\Phi^{CLS} \in \mathbb{R}^{1 \times d_1}$ represents the class tokens of these features, and $\Phi^{Full} \in \mathbb{R}^{g \times d_1}$ represents their full tokens, where $g$ denotes the sequence length and $d_1$ denotes the feature dimension.

HMoE-FFM first deploys layer-specific expert networks $Expert_l$—each comprising a linear layer with GELU (Hendrycks & Gimpel, 2016) activation and layer normalization—to process the full tokens of hierarchical features, formalized as follows:

$$e_l = Expert_l(\Phi_l^{Full}), \quad l \in [Low, Mid, High]. \tag{6}$$

Additionally, a routing module (a two-layer Multi-Layer Perceptron with Tanh as the middle activation function) dynamically calibrates the fusion coefficients for each expert's output based on the class tokens of hierarchical features:

$$w_l = Route_l(\Phi_l^{CLS}), \quad l \in [Low, Mid, High], \quad \sum_l w_l = 1. \tag{7}$$

Finally, the output feature is the weighted fusion of all experts' outputs:

$$\Phi_{Fused} = \sum_l w_l \cdot e_l, \quad l \in [Low, Mid, High]. \tag{8}$$

Notably, this approach not only leverages the complementary strengths of multi-level features to preserve both fine-grained visual details and semantic consistency, but also enables precise modulation of visual granularity. For instance, users can manually adjust the fusion coefficients (Eq. 7) of

the experts' outputs, thereby achieving flexible control over the granularity of concept preservation tailored to specific needs—as illustrated in Fig. 1 (b) and more results in Sec. B.4. Comparisons between our HMoE-FFM and other widely used feature fusion approaches will be presented in later Sec. 5.4. It is worth emphasizing that while our method is instantiated on the CLIP model in this work, *similar observations and conclusions may be generalized to other image encoders—a direction we reserve for future exploration.*

### 4.3 MULTI-SUBJECT PERSONALIZATION

*Without requiring additional training on multi-subject datasets*, our DynaIP can be directly extended to multi-subject personalization scenarios via a straightforward mask-guided region-level feature injection. Concretely, given $N$ reference images corresponding $N$ masks, we first compute the CA between noisy image tokens $X$ and each reference image tokens $C_i$ as specified in Eq. (2). These CA outputs are then added to the corresponding regions of the text CA output features, guided by their respective masks:

$$DCA''(T, X, C) = X^{MMA} + \sum_{i=1}^{N} \lambda_i \cdot M_i \cdot CA(X, C_i), \qquad (9)$$

where $X^{MMA}$ implies the output of text CA as defined in Eq. (3), $M_i$ is a binary mask specifying the regions in the generated image where the subject needs to be replaced by the subject from the $i_{th}$ reference image; it can either be manually annotated by users or automatically generated using existing grounding and segmentation tools (Liu et al., 2024; Kirillov et al., 2023). In our experiments, we adopt the automatic approach or preset masks for multi-subject personalization. $\lambda_i$ is the weight factor to regulate the influence of the $i_{th}$ reference image features. As will be shown in later Sec. 5.4, thanks to our proposed Dynamic Decoupling Strategy, our results can significantly reduce inconsistencies when composing different reference subjects and greatly enhance the overall harmony of multi-subject generation—even when these subjects possess distinct styles and visual attributes. Furthermore, this method facilitates effective object replacement, which in turn supports flexible editing of the generated results, as shown in the last case of Fig. 1 (c).

## 5 EXPERIMENTAL RESULTS

### 5.1 EXPERIMENT SETTINGS

**Implementation Details.** Please refer to Sec. B.1 for implementation details.

**Datasets.** Please refer to Sec. B.1 for details of training datasets. For evaluation, we assess SS-PT2I performance on DreamBench++ (Peng et al., 2025), which contains 1350 test samples across various categories—including animals, humans, and objects—and covers both photorealistic and non-photorealistic styles. For MS-PT2I evaluation, we follow previous studies (Wang et al., 2025; Huang et al., 2025) to establish a new benchmark named DynaIP-Bench. Specifically, we source subjects from existing works (Peng et al., 2025; Wang et al., 2025; Chen et al., 2025) to construct 642 two-subject pairs and 246 three-subject triplets, forming a comprehensive set of 888 multi-subject test samples. More details are provided in Sec. B.1.

**Evaluation Metrics.** Previous works (Ruiz et al., 2023; Wang et al., 2025) typically compare feature-based similarity metrics from models like CLIP (Radford et al., 2021) and DINO (Oquab et al., 2024). However, as highlighted in (Peng et al., 2025; Cai et al., 2025), these metrics only capture global semantic similarity and suffer from extreme noise and bias. Therefore, we adhere to the evaluation protocol outlined in (Peng et al., 2025), which systematically assesses PT2I performance through two key dimensions: Concept Preservation (CP) and Prompt Following (PF), using a Vision-Language Model (VLM) (Hurst et al., 2024). This protocol demonstrates better alignment with human preferences compared to traditional metrics. For reproducibility and mitigating model-specific biases, we employ two SOTA open-source VLMs—InternVL3-78B (Zhu et al., 2025) and Qwen3-VL-32B (Team, 2025)—and report the average of their scores. The goal of the PT2I task is to maximize the Nash utility (Nash et al., 1950), defined as the product of CP and PF scores. Note that for MS-PT2I scenarios, we use the average CP score across all subjects for evaluation. The detailed VLM evaluation prompts can be found in Sec. B.1.

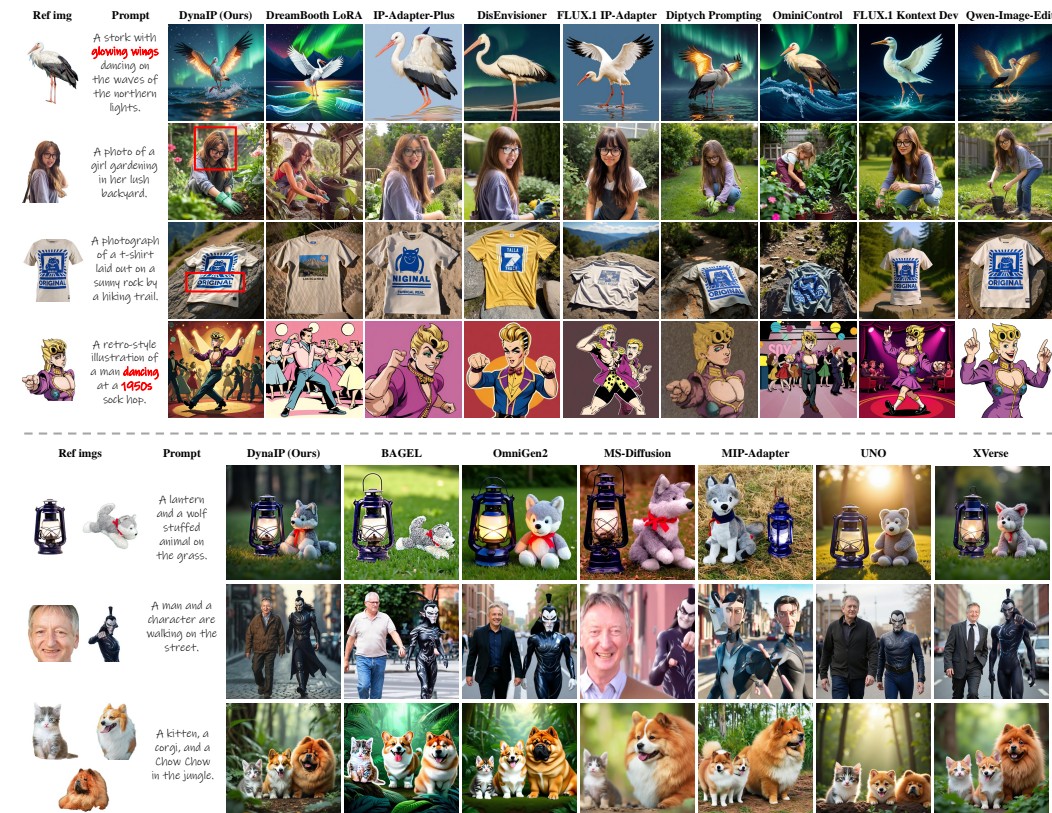

Figure 5: **Qualitative comparisons** on single- (**top**) and multi-subject (**bottom**) PT2I generation.

**Compared Baselines.** For SS-PT2I, we compare our model with leading open-source methods such as finetuning-based DreamBooth (Ruiz et al., 2023), DreamBooth LoRA (Hu et al.), and Textual Inversion (Gal et al., 2023); zero-shot UNet-based IP-Adapter-Plus (Ye et al., 2023) and DisEnvisioner (He et al., 2025); DiT-based FLUX.1 IP-Adapter (Team, 2024), Diptych Prompting (Shin et al., 2024), OminiControl (Tan et al., 2025), Self-Distillation (Cai et al., 2025), and FLUX.1 Kontext Dev (Labs et al., 2025); MLLM-based Qwen-Image-Edit (Wu et al., 2025a). For MS-PT2I, we compare our model with MLLM-based BAGEL (Deng et al., 2025) and Omni-Gen2 (Wu et al., 2025b), and other finetuning-free methods like MS-Diffusion (Wang et al., 2025), MIP-Adapter (Huang et al., 2025), UNO (Wu et al., 2025c), and XVerse (Chen et al., 2025). For fair comparison, we adapt input prompts to the officially recommended format of each compared method to avoid biases from mismatched prompt formats.

## 5.2 QUALITATIVE COMPARISON

We present qualitative comparison results for SS-PT2I in Fig. 5 (top). As illustrated, our DynaIP preserves fine-grained subject details with high fidelity—such as the facial features in Row 2 and the legible text in Row 3—while simultaneously achieving strong prompt alignment, as evidenced by the glowing wings in Row 1 and the 1950s-style dancing in Row 4. In contrast, existing approaches either fail to maintain fine-grained subject consistency or struggle to adhere to input prompts.

Furthermore, we showcase qualitative comparisons for MS-PT2I in Fig. 5 (bottom). Notably, while all competing methods are trained on carefully curated multi-subject datasets, our DynaIP delivers superior performance in both subject consistency and visual harmony—*despite being trained exclusively on single-subject data*. Specifically, our method generates plausible, harmonious, and naturally coordinated compositions, even when reference subjects exhibit distinct styles and visual attributes (e.g., Row 2). By contrast, existing approaches frequently produce copy-paste artifacts or inconsistent subject renderings, resulting in reduced concept fidelity and visually disjointed compositions. More qualitative results are provided in Sec. B.2.

Table 1: **Quantitative comparisons** with SOTA methods. CP: Concept Preservation, PF: Prompt Following. The  highest  and  second-highest  scores are highlighted.

| Method | Single-subject | | | Multi-subject | | |
| --- | --- | --- | --- | --- | --- | --- |
| | CP | PF | CP · PF | CP | PF | CP · PF |
| DreamBooth | 0.458 | 0.721 | 0.330 | - | - | - |
| DreamBooth LoRA | 0.594 | 0.840 | 0.499 | - | - | - |
| Textual Inversion | 0.384 | 0.633 | 0.220 | - | - | - |
| IP-Adapter-Plus | **0.738** | 0.668 | 0.493 | - | - | - |
| DisEnvisioner | 0.559 | 0.664 | 0.371 | - | - | - |
| FLUX.1 IP-Adapter | 0.681 | 0.600 | 0.408 | - | - | - |
| Diptych Prompting | 0.616 | 0.839 | 0.517 | - | - | - |
| OminiControl | 0.596 | 0.895 | 0.534 | - | - | - |
| Self-Distillation | 0.513 | 0.870 | 0.447 | - | - | - |
| FLUX.1 Kontext Dev | 0.718 | 0.893 | 0.641 | - | - | - |
| Qwen-Image-Edit | 0.693 | 0.928 | 0.643 | - | - | - |
| BAGEL | 0.603 | 0.926 | 0.558 | 0.566 | 0.885 | 0.500 |
| OmniGen2 | 0.622 | 0.922 | 0.574 | 0.552 | 0.952 | 0.526 |
| MS-Diffusion | 0.686 | 0.828 | 0.568 | 0.584 | 0.850 | 0.496 |
| MIP-Adapter | 0.692 | 0.649 | 0.449 | 0.388 | 0.713 | 0.276 |
| UNO | 0.721 | 0.799 | 0.576 | 0.509 | 0.857 | 0.436 |
| XVerse | 0.643 | 0.869 | 0.559 | 0.548 | 0.890 | 0.488 |
| **Our DynaIP** | 0.696 | **0.934** | **0.650** | **0.617** | **0.997** | **0.615** |

## 5.3 QUANTITATIVE COMPARISON

**Automatic Scores.** Quantitative comparison results for both SS- and MS-PT2I are presented in Tab. 1, where we report VLM evaluation metrics following the protocol in (Peng et al., 2025). For SS-PT2I, our method achieves the highest PF and CP·PF scores. The PF score quantifies adherence to text prompts, while the CP·PF score reflects balanced performance between subject consistency and prompt alignment. As explicitly emphasized in (Peng et al., 2025), this balanced CP·PF performance is the ultimate objective of the PT2I task. Although our CP score is marginally lower than that of certain competing methods (*e.g.*, IP-Adapter-Plus (Ye et al., 2023)), we attribute this discrepancy primarily to two factors: first, these baselines fail to effectively decouple concept-agnostic information, leading generated images closely resemble the input without meaningful transformation, as illustrated in Fig. 5 and further validated by their substantially lower PF scores; second, the CP score itself exhibits limited sensitivity to both such copy-paste artifacts and fine-grained subject details. Therefore, to more effectively demonstrate the superiority of our DynaIP in terms of CP, we further conduct additional user studies in Sec. B.5, where our method outperforms all competitors.

For MS-PT2I, our method outperforms all baselines by achieving the highest scores across all metrics, demonstrating its superiority in both maintaining subject consistency and prompt adherence. Notably, our CP score in MS scenarios surpasses those of all competitors—a stark contrast to the SS scenarios. We hypothesize this is because existing approaches frequently suffer from critical flaws such as subject omission, identity confusion, or unnatural subject fusion—issues that directly degrade the performance of CP, as shown in Fig. 5. In contrast, our method effectively mitigates these problems and generates visually harmonious multi-subject compositions, thanks to our proposed Dynamic Decoupling Strategy and mask-guided feature injection.

## 5.4 ABLATION STUDIES

**Effects of Dynamic Decoupling Strategy (DDS).** We conduct ablation experiments on our proposed DDS. As shown in Fig. 6-left, without DDS, the concept-agnostic information of the reference images, such as posture, perspective, and illumination, is retained in the generated results, leading to degraded editability and disharmonious multi-subject compositions. These issues are effectively mitigated by DDS, confirming its ability to disentangle concept-specific information from concept-agnostic information. Additionally, quantitative results in Tab. 2 (1-2) further validate its effectiveness in enhancing the CP·PF balance and multi-subject scalability. More results and analyses can be found in Sec. B.6.

**Superiority of HMoE-FFM.** We compare our HMoE-FFM with two widely adopted feature fusion methods: element-wise addition (add) and channel-wise concatenation (concat). As illustrated in

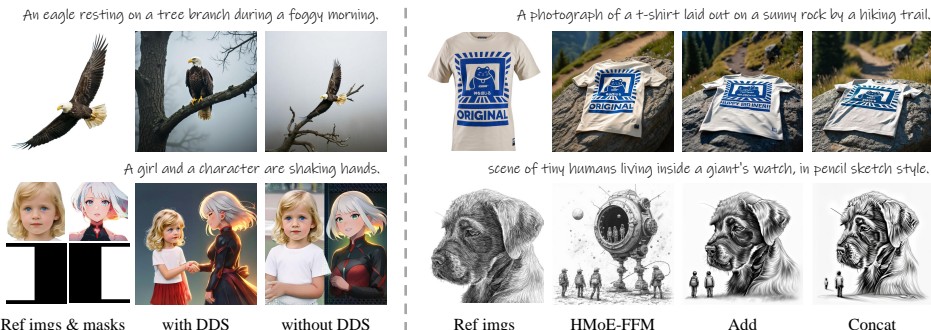

Figure 6: **Qualitative ablation study results** of **Left**: Dynamic Decoupling Strategy (DDS) and **Right**: different feature fusion approaches.

Table 2: **Quantitative ablation study results.** CP: Concept Preservation, PF: Prompt Following. The highest and second-highest scores are highlighted.

| Setting | Single-subject | | | Multi-subject | | |
|---|---|---|---|---|---|---|
| | CP | PF | CP · PF | CP | PF | CP · PF |
| (1) **Full Model** | 0.696 | 0.934 | **0.650** | **0.617** | **0.997** | **0.615** |
| (2) w/o DDS | **0.785** | 0.799 | 0.627 | 0.499 | 0.545 | 0.272 |
| (3) add fusion | 0.693 | 0.910 | 0.631 | 0.612 | 0.994 | 0.608 |
| (4) concat fusion | 0.691 | 0.903 | 0.624 | 0.605 | 0.992 | 0.600 |
| (5) only shallow-layer | 0.627 | 0.924 | 0.579 | 0.464 | 0.991 | 0.460 |
| (6) only mid-layer | 0.670 | 0.928 | 0.622 | 0.603 | 0.993 | 0.599 |
| (7) only deep-layer | 0.480 | **0.950** | 0.456 | 0.474 | 0.995 | 0.471 |

Fig. 6-right and Tab. 2 (3-4), fusing multi-layer CLIP features via add or concat yields suboptimal performance on both CP and PF—particularly in terms of fine-grained fidelity and style transfer. This highlights the superiority of our HMoE-FFM, which more effectively integrates hierarchical features through layer-specific expert networks and a dynamic routing mechanism that adapts to the characteristics of each reference image. Additionally, as shown in Fig. 1 (b) and Sec. B.4, our HMoE-FFM provides greater flexibility to enable run-time control over the granularity of concept preservation, a capability unattainable with add and concat approaches. On the other hand, to complement Fig. 4-right, we further present quantitative results for HMoE-FFM using different CLIP layers individually in Tab. 2 (5-7). Evidently, while shallow-layer features excel at capturing low-level patterns (e.g., lines and text), their limited capacity to grasp mid- and high-level attributes results in lower CP scores. Similarly, deep-layer features yield lower CP scores due to their lack of low-level and fine-grained information; however, they achieve the highest PF scores by virtue of their stronger ability to capture semantic information. In contrast, mid-layer features primarily focus on fine-grained details while incorporating a certain amount of semantic information, thus striking a more favorable balance between CP and PF. By harnessing the strengths of these multi-layer features, our HMoE-FFM integrates low-level, fine-grained, and semantic information, thereby achieving an optimal balance between CP and PF. More results and analyses can be found in Sec. B.7.

## 6 CONCLUSION

In this paper, we present DynaIP, a plug-and-play adapter to empower diffusion transformers for scalable zero-shot personalized text-to-image generation. Our DynaIP introduces two key innovations. The first is a *Dynamic Decoupling Strategy (DDS)* to disentangle concept-specific information from concept-agnostic information within reference images. This strategy significantly enhances the critical balance between concept preservation and prompt following, while simultaneously bolstering the scalability of multi-subject compositions. The second is a *Hierarchical Mixture-of-Experts Feature Fusion Module (HMoE-FFM)* to dynamically integrates the multi-level CLIP features to preserve both fine-grained visual details and semantic consistency. It not only enhances the concept fidelity of reference images but also provides flexible control over visual granularity. Extensive experiments show the superiority of our DynaIP in both single- and multi-subject personalization, while *requiring only single-subject training datasets*.

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

# A  RELATED WORK

## A.1  TEXT-TO-IMAGE GENERATION

Text-to-Image (T2I) generative models have experienced explosive growth in recent years. While some research endeavors employ Generative Adversarial Networks (GANs) (Reed et al., 2016; Kang et al., 2023) or autoregressive (Ding et al., 2021; Ramesh et al., 2021; Yu et al., 2022) paradigms, the majority of contemporary T2I frameworks opt for denoising diffusion models (Ho et al., 2020; Esser et al., 2024) owing to their notable advantages in generation quality. Early pioneering models, including LDM (Rombach et al., 2022), DALL-E 2 (Ramesh et al., 2022), Imagen (Saharia et al., 2022), and SDXL (Podell et al.), commonly utilized U-Net (Ronneberger et al., 2015) for noise prediction. Recent studies have replaced the traditional U-Net with scalable transformer (Vaswani et al., 2017) architectures, giving rise to more advanced models such as Diffusion Transformers (DiT) (Peebles & Xie, 2023; Chen et al., 2024c). Subsequent works, such as Stable Diffusion 3 (Esser et al., 2024) and FLUX.1 (Labs, 2024), have further extended DiT to multimodal DiT (MM-DiT), achieving SOTA T2I generation performance. In this work, we build our methodology on MM-DiT, but *the main ideas and modules may also be applicable to other architectures*.

## A.2  PERSONALIZED TEXT-TO-IMAGE GENERATION

Personalized Text-to-Image (PT2I) generation has garnered significant attention in both academia and industry. Early research predominantly relied on finetuning-based methods (Kumari et al., 2023; Liu et al., 2023; Gu et al., 2023; Jiang et al., 2024). For instance, DreamBooth (Ruiz et al., 2023) and Textual Inversion (Gal et al., 2023) bound visual concepts to text identifiers via finetuning, while LoRA (Hu et al.) introduced lightweight parameters to enable efficient adjustments. However, these methods are plagued by the heavy burden of per-subject finetuning. In response, IP-Adapter (Ye et al., 2023) and BLIP Diffusion (Li et al., 2023) incorporated additional image encoders and layers to process reference images and injecting image features into diffusion models. These adapter-based techniques enable zero-shot PT2I generation, emerging as the dominant research paradigm and inspiring a wealth of follow-up works (Wei et al., 2023; Chen et al., 2024d; Wang et al., 2024b;a; Guo et al., 2024; Huang et al., 2024b; He & Yao, 2025; Kong et al., 2025), including several multi-subject approaches (Xiao et al., 2024a; Wei et al., 2024; Wang et al., 2025; Huang et al., 2025; Ma et al., 2024; Zhang et al., 2024). Aligning with part of our motivations, (He et al., 2025) proposed DisEnvisioner, which disentangles and enriches concept-specific attributes by learning individual image tokens. In contrast, our method directly leverages the decoupled learning behavior of the dual branches in MM-DiT to separate concept-specific information from concept-agnostic content. Furthermore, DisEnvisioner still suffers from the loss of fine-grained concept details—largely because it relies on deep-layer features from CLIP, as illustrated in Fig. 2 (b). Additionally, most of the aforementioned efforts are built on U-Net-based diffusion models, with limited exploration of adapter-based PT2I techniques for SOTA DiT-based models such as FLUX.1 (Labs, 2024).

For DiT architectures, methods such as OminiControl (Tan et al., 2025) adopt unified conditioning strategies to handle text embeddings, latent tokens, and VAE-encoded reference subjects. Although showing promise in PT2I generation (Shin et al., 2024; Cai et al., 2025; Mao et al., 2025; Huang et al., 2024a; Wu et al., 2025c; Mou et al., 2025; Guo et al., 2025; Chen et al., 2025), they often necessitate constructing complex cross-pair and multi-subject datasets for base model finetuning. In addition, the full attention mechanism's computational complexity escalates exponentially with more reference conditions, particularly in multi-subject scenarios, resulting in lower-resolution outputs and limited flexibility compared to adapter-based methods.

Another category of methods leverages Multimodal Large Language Model (MLLM) (Zhu et al., 2024; Bai et al., 2025) to conduct multi-modal training through text-image interleaving to bridge the gap between image and text prompts (Sun et al., 2024; Pan et al.; Li et al., 2024; Patel et al.; Xiao et al., 2024b; Wu et al., 2025b; Deng et al., 2025; Wu et al., 2025a). Yet, these methods typically require re-training MLLM encoders and generators, imposing substantial demands on training resources.

### A.3 MULTI-LAYER FEATURE FUSION

In the context of MLLMs, several empirical studies (Chen et al., 2024b; Yao et al., 2024; Cao et al., 2024) have shown that multi-layer visual features can enhance model performance. To systematically explore the integration of multi-layer visual features in MLLMs, (Lin et al., 2025) categorizes existing fusion strategies into four distinct categories and reveals that direct fusion (addition or concatenation) at the input stage consistently yields superior performance across various configurations. However, due to the fundamental task divergence between understanding-oriented and generation-oriented tasks, the observations derived from MLLM research may not be generalizable to our PT2I scenario (see comparison results in Sec. 5.4). For the PT2I tasks, while a handful of works (Wei et al., 2023; Zhang et al., 2024) have also attempted to fuse multi-layer features for performance improvement, their fusion strategies remain limited to simple addition or concatenation. In contrast, our HMoE-FFM leverages a MoE architecture to integrate hierarchical visual features, with a dynamic routing mechanism that adaptively calibrates fusion coefficients based on the unique attributes of each input reference image. This approach not only enhances fine-grained visual details and semantic consistency but also facilitates precise modulation of visual granularity, as validated in Secs. 5.4, B.4, and B.7.

## B MORE EXPERIMENTAL RESULTS

### B.1 MORE EXPERIMENT SETTINGS

**Implementation Details.** We build our model based on FLUX.1-Dev (Labs, 2024), with OpenCLIP ViT-L/14-336 adopted as the image encoder. The FLUX.1-Dev architecture includes 57 MM-DiT blocks; we augment each block with a new image cross-attention layer. For the HMoE-FFM, we utilize features from CLIP's layer 10, 17, and 24 as the input features for the low-, mid-, and high-level expert networks, respectively (see ablation results across more layers in Sec. B.8). The total number of trainable parameters in our DynaIP is approximately 1.4B. During training, we only optimize the augmented layers while keeping the parameters of the pretrained models fixed. The training objective is the flow-matching loss, which is consistent with the original FLUX.1-Dev base model. The training process is divided into two sequential stages: The initial stage (20,000 iterations) establishes fundamental capabilities through exclusive intra-pair training, developing robust subject-specific adaptation. Building upon this foundation, the second stage (80,000 iterations) leverages the cross-pair dataset to mitigate copy-paste artifacts (Wang et al., 2025). For optimization, we employ the AdamW optimizer with a cosine learning rate scheduler initialized at 0.00002, combined with a weight decay of 0.0001. All experiments were conducted on 8 Ascend 910B NPUs, with a total batch size of 16 and a training resolution of $1024 \times 1024$. To enable classifier-free guidance, we use a 5% probability to drop text and image individually, and a 5% probability to drop text and image simultaneously. For the training of HMoE-FFM, each expert's feature is dropped independently with a 5% probability. Finally, we set the image weight factor $\lambda$ to 1.0 for both training and inference stages.

**Training Datasets.** We design two specialized datasets to support our two-stage training process. For the first training stage, we construct an intra-pair dataset containing around 0.3M images filtered from several open-source datasets, including FFHQ-wild (Karras et al., 2019) and SA-1B (Kirillov et al., 2023). To decouple background information from the target subject, we use off-the-shelf object detection (Liu et al., 2024) and segmentation (Kirillov et al., 2023) techniques to extract the primary subject, and then replace the original background with a uniform white color. For human subjects, we randomly extract either faces or entire bodies. The resulting segmented image is designated as the reference image, while the original image serves as the ground truth for training. For the second training stage, drawing inspiration from (Wang et al., 2025; Tan et al., 2025), we construct a cross-pair dataset with roughly 1M images from open-source datasets—including VITON-HD (Choi et al., 2021), AnyInsertion (Song et al., 2025), and OpenS2V-Nexus (Yuan et al., 2025)—as well as images generated by FLUX.1-Dev (Labs, 2024). Similar to the intra-pair dataset, all reference images here undergo the same object detection and segmentation pipeline described above. *Note that every image in both training datasets contains only one primary subject, and our model has not been trained on multi-subject datasets.*

Table 3: **Prompt details of our multi-subject DynaIP-Bench.** Each combination type has preset prompts. [P] denotes prompt variations about the scene or actions.

| Type | Prompt | [P] |
|---|---|---|
| living+living
living+object
object+object | a {0} and a {1} [P],
{0} on the left, and
{1} on the right | in a room
in the snow
in the jungle
on the beach
on the grass
on a cobblestone street |
| human+human
human+character
character+character | a {0} and a {1} [P],
{0} on the left, and
{1} on the right | embracing
are sitting on the sofa
are eating noodles
are shaking hands
are fighting
are walking on the street |
| living+upwearing
living+midwearing
living+wholewearing | a {0} wearing a {1} [P] | in a room
in the snow
in the jungle
on the beach
on the grass
on a cobblestone street |
| midwearing+downwearing | a woman wearing a {0}
and {1} [P] | in a room
in the snow
in the jungle
on the beach
on the grass
on a cobblestone street |
| living+living+living
object+object+object | a {0}, a {1}, and a {2} [P],
{0} on the left,
{1} on the middle, and
{2} on the right | in a room
in the snow
in the jungle
on the beach
on the grass
on a cobblestone street |
| human+human+character | a {0}, a {1}, and a {2} [P],
{0} on the left,
{1} on the middle, and
{2} on the right | are sitting on the sofa
are eating noodles
are fighting
are walking on the street
are taking a photo together
are playing football |
| upwearing+midwearing+
downwearing | a woman wearing a {0}, a
{1}, and a {2} [P] | in a room
in the snow
in the jungle
on the beach
on the grass
on a cobblestone street |

**Details of DynaIP-Bench.** Following prior studies (Wang et al., 2025; Huang et al., 2025), we construct our multi-subject DynaIP-Bench by sourcing unseen subjects from existing works (Peng et al., 2025; Wang et al., 2025; Chen et al., 2025). Our DynaIP-Bench comprises 8 data types and 14 combination types involving two or three subjects, encompassing animals, humans, objects, and clothing, and covers both photorealistic and non-photorealistic styles. We provide the details in Tab. 3. Each combination type has 6 prompt variations. We randomly generate a total of 888 combinations, consisting of 642 two-subject combinations and 246 three-subject combinations. Compared to other multi-subject benchmarks, our DynaIP-Bench ensures that model performance is comprehensively reflected across a rich set of cases with diverse types and styles. Notably, we explicitly incorporate multi-subject interaction scenarios (e.g., embracing, shaking hands, and fighting), which not only increase the test set's complexity but also enable targeted assessment of different models' capabilities to generate natural and logical multi-subject interactions—a core requirement for practical personalized text-to-image generation.

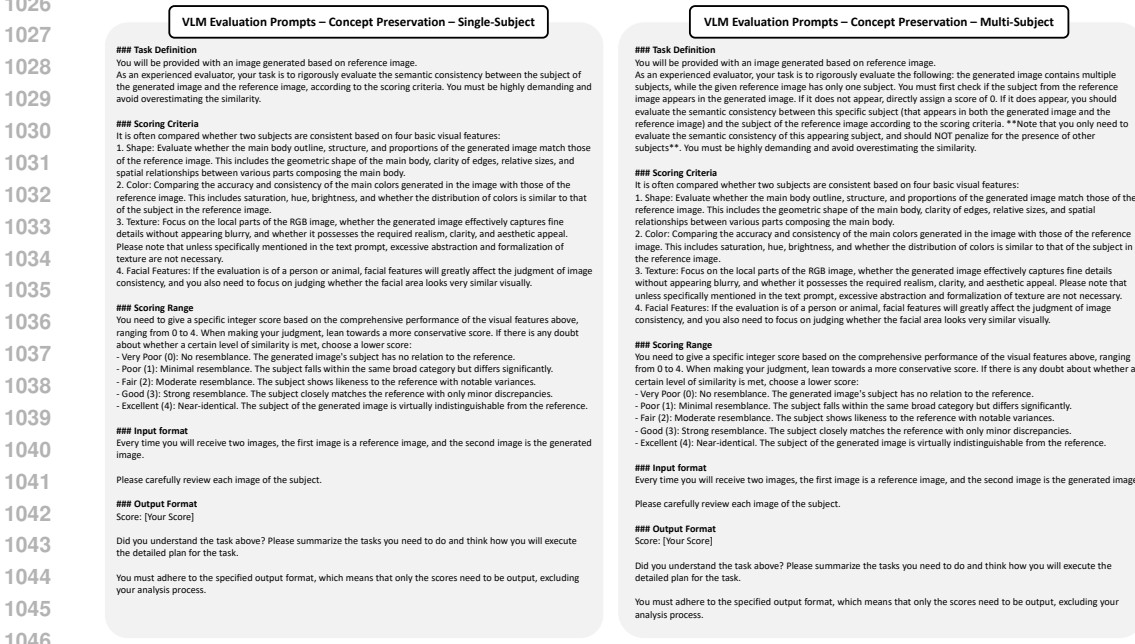

Figure 7: **VLM evaluation prompts for Concept Preservation (CP)** on **Left**: single-subject and **Right**: multi-subject PT2I generation tasks.

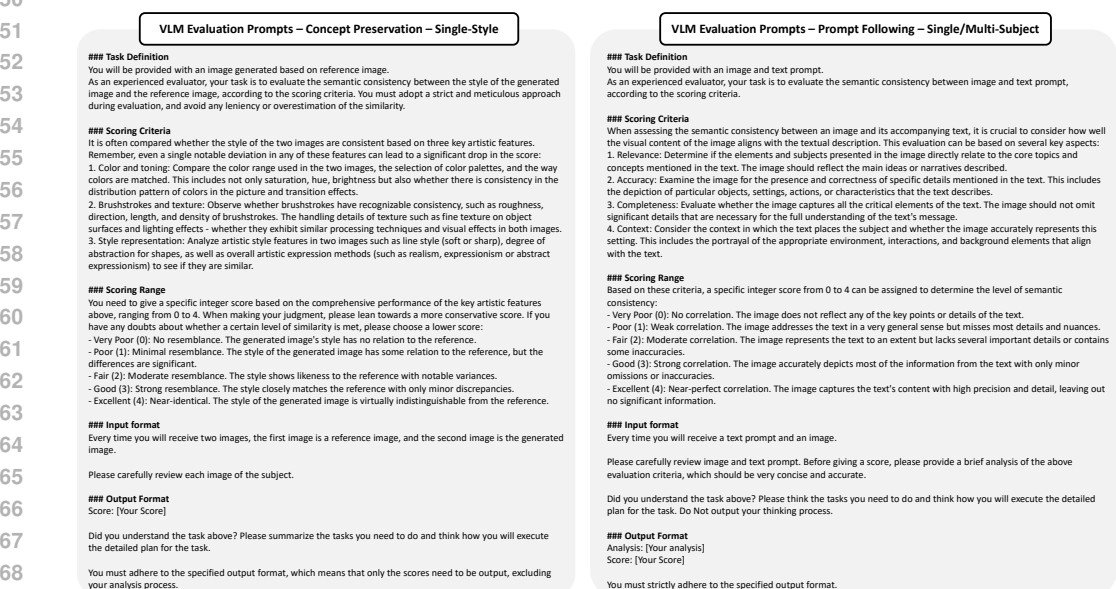

Figure 8: **VLM evaluation prompts** for **Left**: **Concept Preservation (CP)** on **single-style** PT2I generation tasks and for **Right**: **Prompt Following (PF)** on both single- and multi-subject PT2I generation tasks.

**Details of Evaluation Metrics.** We adhere to the evaluation protocol outlined in (Peng et al., 2025), which systematically assesses PT2I performance through two core metrics: Concept Preservation (CP) and Prompt Following (PF), using a Vision-Language Model (VLM) (Hurst et al., 2024). Detailed VLM evaluation prompts are presented in Figs. 7 and 8; these prompts are largely consistent with those in (Peng et al., 2025), with only minor modifications tailored to MS-PT2I scenarios (Fig. 7-Right). Specifically, Concept Preservation (CP) quantifies the visual consistency between the sub-

jects of generated images and those of reference images. For MS-PT2I scenarios, we independently evaluate the CP score for each reference subject and compute the average CP score across all subjects to ensure assessment comprehensiveness. Additionally, (Peng et al., 2025) employs a distinct prompt for evaluating style-related CP (Fig. 8-Left); we retain this prompt to assess style-guided generation within the DreamBench++ (Peng et al., 2025). On the other hand, Prompt Following (PF) measures the semantic consistency between generated images and input text prompts. Notably, the PF evaluation prompt remains identical for both SS- and MS-PT2I tasks, as detailed in Fig. 8-Right.

## B.2 Additional Qualitative Results

Here, we provide additional qualitative comparison results for both single- and multi-subject personalized text-to-image generation, as shown in Figs. 9, 10 and 11. Furthermore, we also present the results of DynaIP in the context of complex interactions between multiple subjects in Sec. B.3.

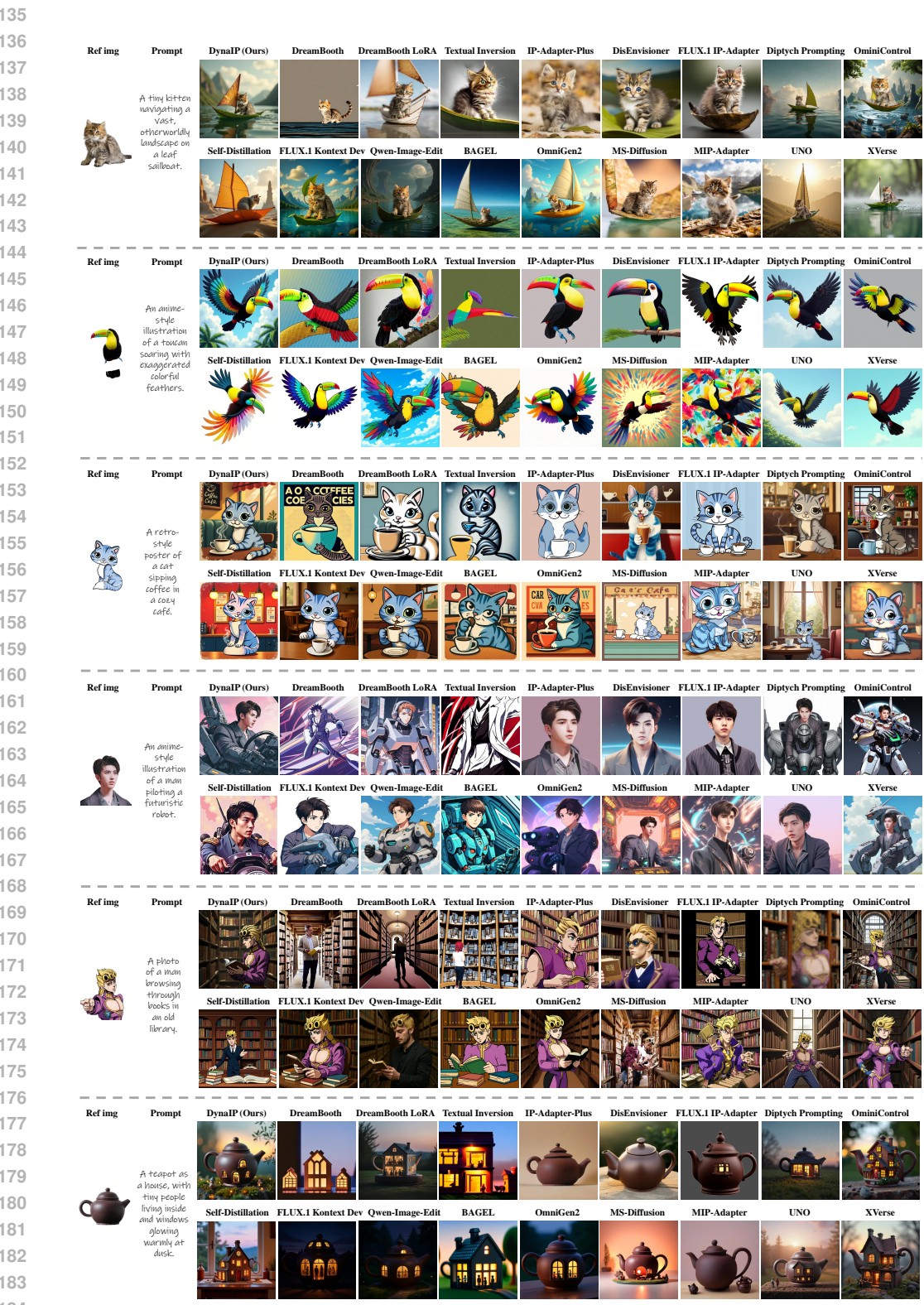

Figure 9: **Additional qualitative comparisons on single-subject PT2I generation.**

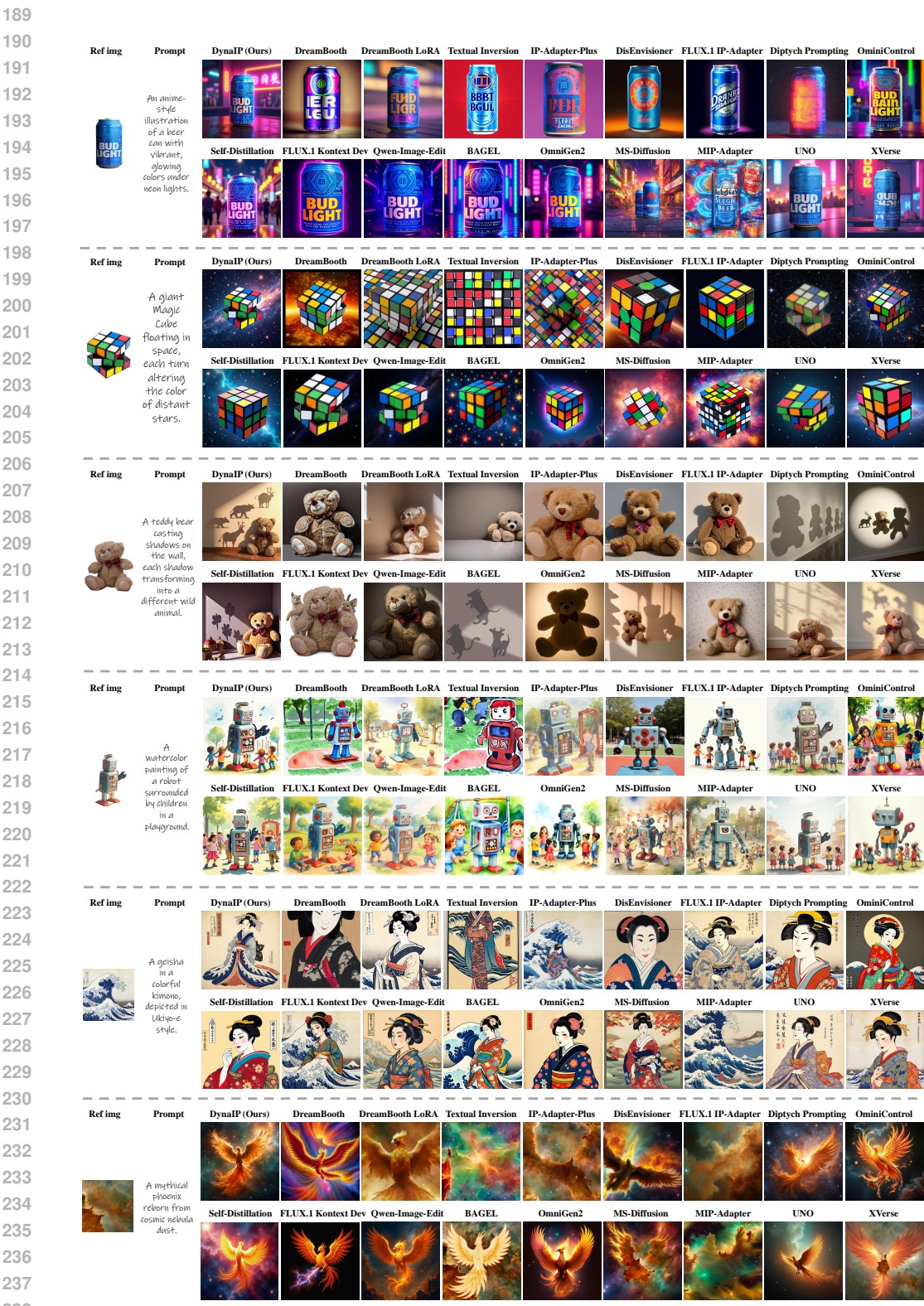

Figure 10: **Additional qualitative comparisons on single-subject PT2I generation.**

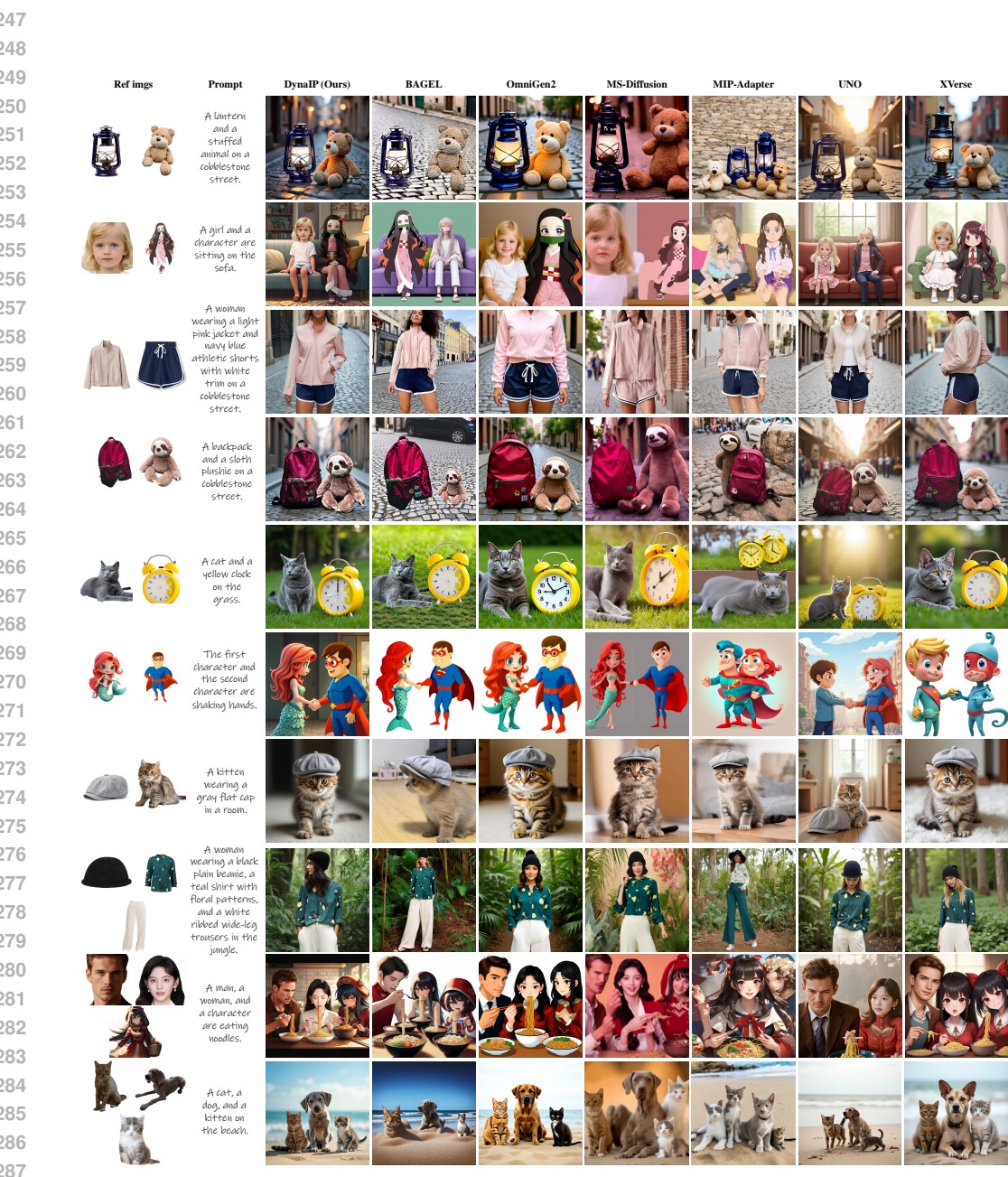

Figure 11: **Additional qualitative comparisons on multi-subject PT2I generation.**

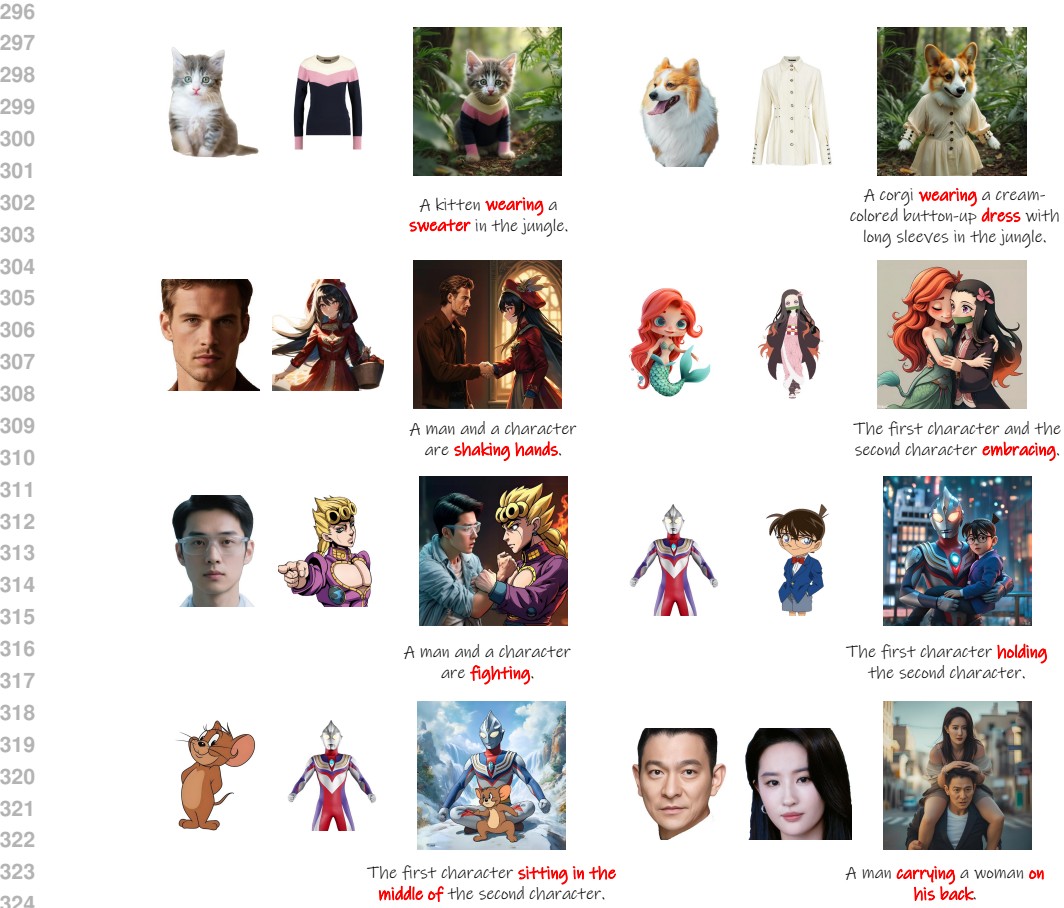

Figure 12: **Exemplar results of prompts with complex interactions of multiple subjects.**

### B.3 MULTI-SUBJECT INTERACTION

The ability to generate natural, logically consistent multi-subject interactions stands as an important requirement for enabling practical, high-quality personalized text-to-image generation. Benefiting from our architecture design and mask-guided feature injection, which preserves the prior of the base model, DynaIP successfully inherits and retains the base model's intrinsic multi-subject interaction capabilities. As illustrated in Fig. 12, DynaIP can flexibly handle and render interactions between reference subjects, even in scenarios where these objects exhibit large spatial overlap. This not only validates its robustness in complex generation tasks but also underscores its strong potential for deployment in practical applications.

### B.4 CONTROL ON VISUAL GRANULARITY

As we introduced in Sec. 4.2, a key advantage of our HMoE-FFM lies in its ability to enable precise modulation of the visual granularity of reference subjects. By allowing users to manually adjust the fusion coefficients (Eq. 7) of experts' outputs, the framework facilitates flexible control over the granularity of concept preservation, tailored to diverse specific needs. To further illustrate this capability, we provide additional examples in Fig. 13 and 14.

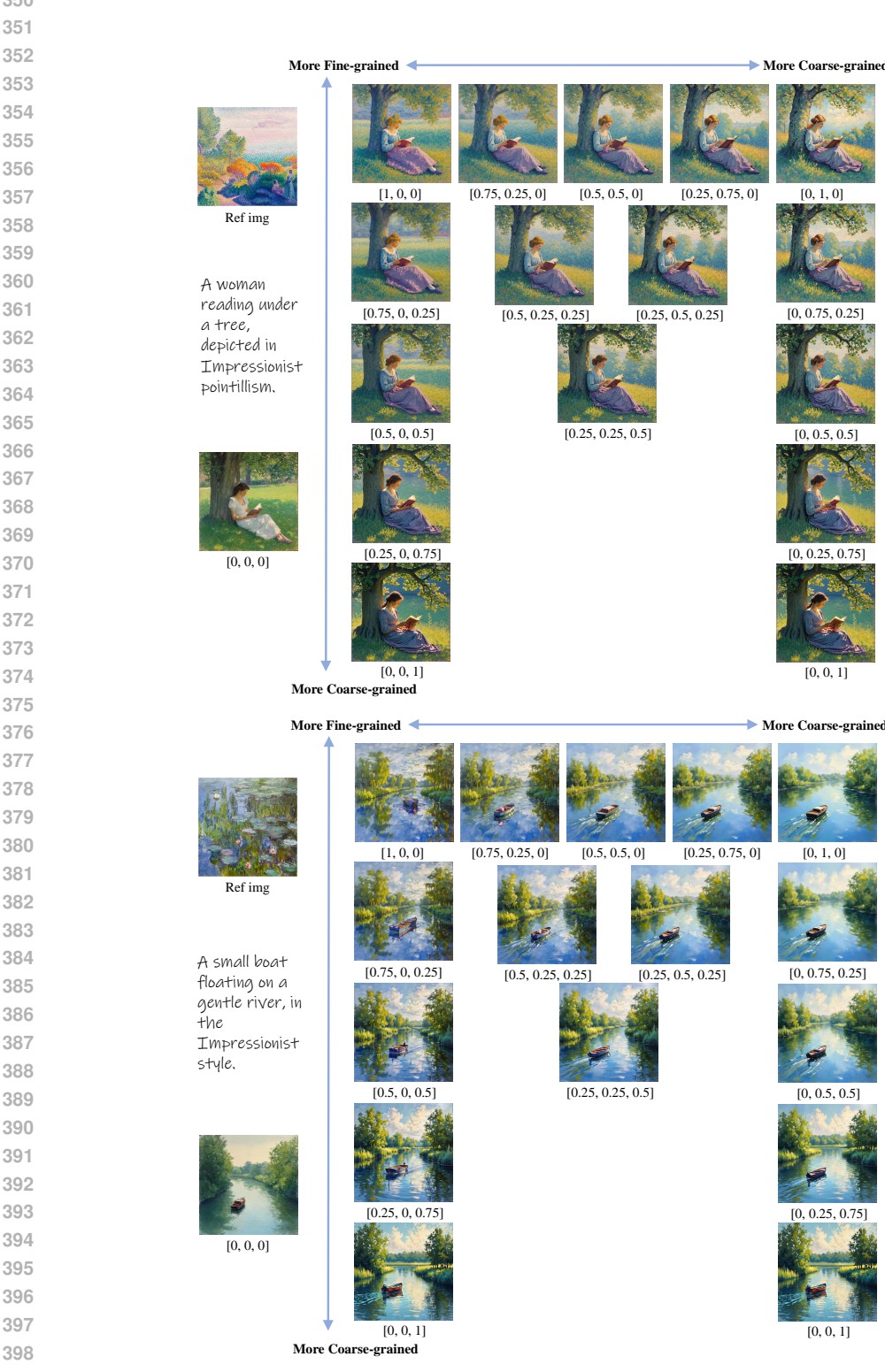

Figure 13: **Control on the visual granularity of concept preservation**, enabled by modulating fusion coefficients ($[w_{Low}, w_{Mid}, w_{High}]$ in Eq. 7) of experts' outputs in HMoE-FFM. Please zoom in to observe the details.

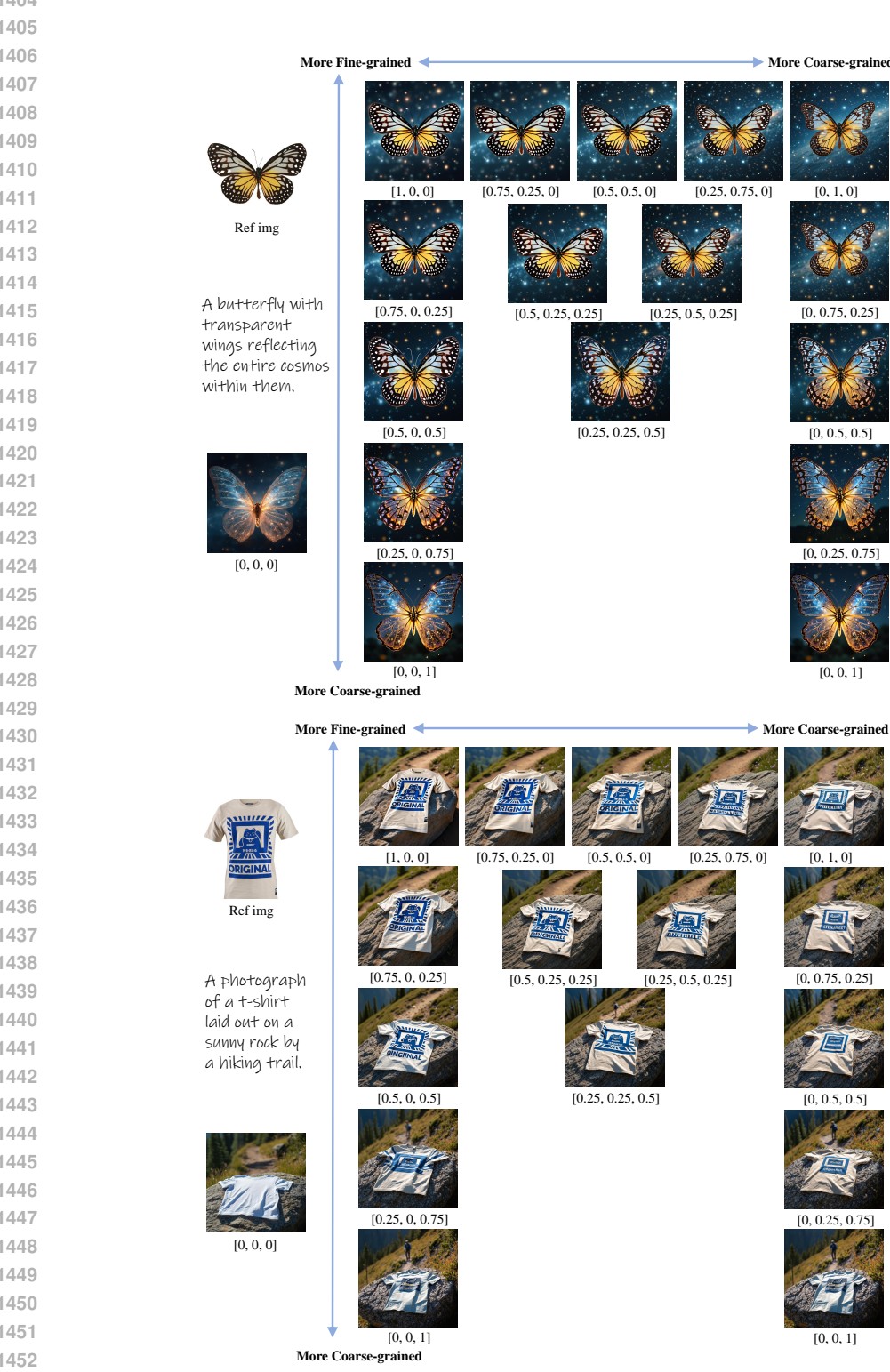

Figure 14: **Control on the visual granularity of concept preservation**, enabled by modulating fusion coefficients ($[w_{Low}, w_{Mid}, w_{High}]$ in Eq. 7) of experts' outputs in HMoE-FFM. Please zoom in to observe the details.

Table 4: **A/B test user study results.** We report the adjusted advantage ratios ($\frac{win+tie}{lose+tie}$) of the competing methods relative to our method. Lower values indicate that our method performs better relative to the competing method. CP: Concept Preservation, PF: Prompt Following. OS: Overall Satisfaction. The **highest** and second-highest scores are highlighted.

| Method | CP | PF | OS |
|---|---|---|---|
| DreamBooth | 0.288 | 0.286 | 0.171 |
| DreamBooth LoRA | 0.709 | 0.782 | 0.660 |
| Textual Inversion | 0.142 | 0.198 | 0.100 |
| IP-Adapter-Plus | 0.745 | 0.442 | 0.297 |
| DisEnvisioner | 0.517 | 0.437 | 0.227 |
| FLUX.1 IP-Adapter | 0.728 | 0.496 | 0.336 |
| Diptych Prompting | 0.742 | 0.726 | 0.568 |
| OmniControl | 0.721 | 0.861 | 0.598 |
| Self-Distillation | 0.644 | 0.737 | 0.521 |
| FLUX.1 Kontext Dev | **0.953** | 0.857 | 0.861 |
| Qwen-Image-Edit | 0.896 | **0.944** | **0.904** |
| BAGEL | 0.718 | 0.802 | 0.622 |
| OmniGen2 | 0.822 | 0.832 | 0.830 |
| MS-Diffusion | 0.647 | 0.487 | 0.368 |
| MIP-Adapter | 0.585 | 0.450 | 0.324 |
| UNO | 0.844 | 0.748 | 0.816 |
| XVerse | 0.825 | 0.786 | 0.828 |

## B.5 USER STUDY

We further conduct A/B test user studies to demonstrate the superiority of DynaIP. For each competing method, we randomly select 30 image combinations per user, encompassing both single-subject and (where applicable) multi-subject personalization tasks. For each combination, we present the results generated by our method and the competing method side by side in a randomized order. Given unlimited time, participants are then instructed to evaluate and select between the two based on three criteria: (1) which result better preserves the reference subject—particularly its fine-grained details (Concept Preservation, CP); (2) which result more accurately aligns with the given prompt (Prompt Following, PF); and (3) which result achieves superior overall satisfaction, including composition quality, naturalness, harmony, and visual appeal (Overall Satisfaction, OS). For each A/B test group, we collect a total of 1,800 votes from 20 users. Tab. 4 reports the adjusted advantage ratios of the competing methods relative to our method—calculated using the formula $\frac{win+tie}{lose+tie}$, where "win" denotes the number of evaluations where the competing method is preferred, "tie" denotes the number of evaluations with no clear preference for either method, and "lose" denotes the number of evaluations where our method is preferred. Lower values indicate that our method performs better relative to the competing method. As the results illustrate, our method achieves better performance than all other competing methods in terms of CP, PF, and OS (all values are lower than 1).

## B.6 MORE ANALYSES OF DYNAMIC DECOUPLING STRATEGY

To better illustrate the decoupling effects of our proposed Dynamic Decoupling Strategy (DDS), we visualize the results generated by injecting reference image features into the MM-DiT under three distinct settings via cross-attentions: (1) text branch only, (2) noisy image branch only, and (3) both text and noisy image branches.

As demonstrated in Fig. 15, compared with the baseline results in column (a), injecting reference image features solely into the text branch of MM-DiT (column (b)) only imparts some concept-agnostic information, such as posture and perspective in the first row, and illumination in the second row. In contrast, injecting reference image features exclusively into the noisy image branch of MM-DiT (column (c)) successfully embeds concept-specific information (e.g., the subject's ID and unique appearance) while effectively disentangling the aforementioned concept-agnostic attributes. The results in column (d) further validate our claims: combining the text branch with the noisy image branch yields outputs where concept-specific and concept-agnostic information are entangled, lead-

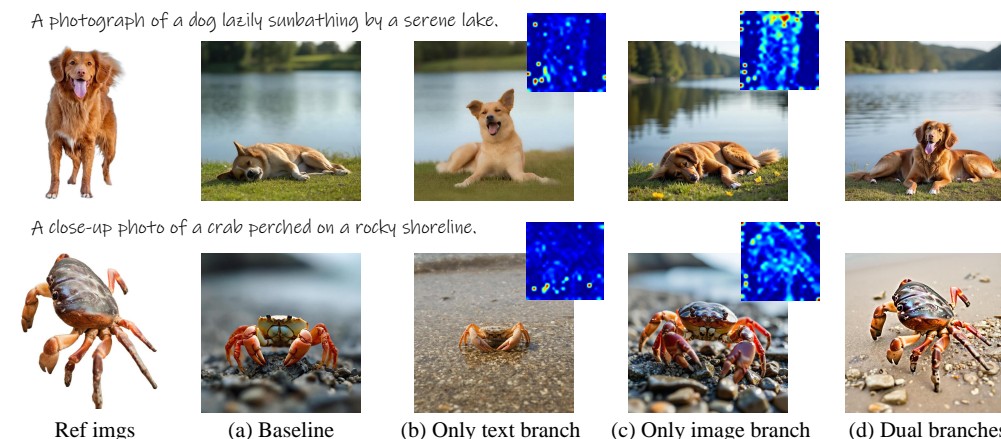

Ref imgs          (a) Baseline          (b) Only text branch          (c) Only image branch          (d) Dual branches

Figure 15: **Detailed analyses of the effects of Dynamic Decoupling Strategy (DDS)**. We visualize the results generated by injecting reference image features into the MM-DiT under three distinct settings via cross attentions: (b) text branch only, (c) noisy image branch only, and (d) both text and noisy image branches. We also present the attention maps illustrating the interaction between each branch's features and the reference image features in the upper right corners of (b) and (c). Please zoom in to observe the details.

ing to prominent copy-paste artifacts that compromise the quality and controllability of generated results.

Additionally, we also visualize the attention maps that illustrate the interaction between each branch's features and the reference image features, which are presented in the upper right corners of columns (b) and (c). As observed, the noisy image branch focuses on the core appearance and semantic regions of the reference subject, whereas the text branch exhibits relatively scattered attention, with higher activation values only in areas irrelevant to the subject's appearance. This observation is well aligned with the phenomena manifested in the generated results.

### B.7    MORE ANALYSES OF HMOE-FFM

We first reveal **how our HMoE-FFM adaptively adjusts the visual granularity of extracted features based on the inherent characteristics of the input reference image**. As illustrated in Fig. 16-Left, we present the fusion coefficients ($[w_{Low}, w_{Mid}, w_{High}]$ in Eq. 7), which are dynamically predicted by the routing module of HMoE-FFM, beneath each input reference image. The coefficient distribution reveals three key patterns: (1) In column (a), for reference images rich in low-level patterns—such as brushstrokes (Row 1), lines (Row 2), and text (Row 3)—the routing module assigns higher fusion coefficients to the low-level expert, followed by the mid-level expert. (2) In column (b), for reference images characterized by fine-grained textures and structures (e.g., facial details in Rows 1 and 2, and surface textures in Row 3), the routing module prioritizes the mid-level expert with higher coefficients, supplemented by the low-level expert. (3) In column (c), for reference images dominated by coarse-grained textures and structures—including anime characters (Row 1) and shape-centric objects (Rows 2 and 3)—the routing module allocates higher fusion coefficients to the high-level and mid-level experts. To summarize, the routing mechanism of HMoE-FFM achieves granularity-adaptive expert selection by dynamically calibrating the fusion weights of low/mid/high-level experts in response to the visual granularity properties of input images. Such granularity-aware routing not only ensures the full utilization of each expert's specialized capabilities but also significantly enhances the model's adaptability to the diverse characteristics of reference images, thereby establishing a robust foundation for high-quality generation in subsequent tasks.

Building on the granularity-aware fused features extracted by HMoE-FFM, we then elucidate **how our model adapts to prompt adjustments across diverse style and content variations**. As demonstrated in Fig. 16-Right, our model achieves superior prompt-following performance, which is

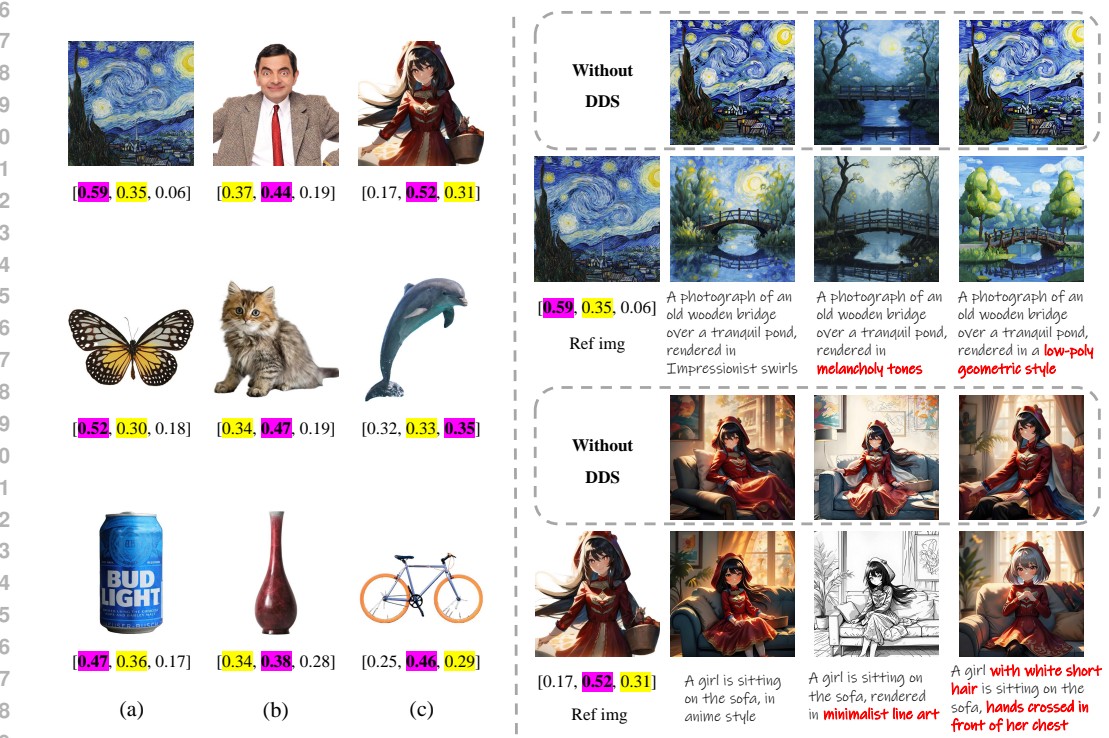

Figure 16: **More analytical results of HMoE-FFM**. **Left:** We present the fusion coefficients ($[w_{Low}, w_{Mid}, w_{High}]$ in Eq. 7), adaptively predicted by the routing module of HMoE-FFM, across diverse input reference images. The **highest** and second-highest coefficients are highlighted. **Right:** we visualize how our model adapts to prompt adjustments across diverse style and content variations.

primarily attributed to the intrinsic properties of Decoupled Cross-Attention (DCA) (Ye et al., 2023) and our proposed Dynamic Decoupling Strategy (DDS). Specifically, the DCA mechanism achieves *prompt-driven attribute selection using noisy (generated) image as a bridge and cross-attention dependency modeling as the theoretical foundation*. It functionally specializes a dual-stream architecture comprising a Text CA stream and an Image CA stream. The Text CA stream first defines the target semantic objective—*i.e.*, which semantic elements (content, style, high-level attributes, or low-level textures) should be prioritized in the noisy image. This capability is inherited from the base T2I model (*e.g.*, FLUX.1-Dev), which is pre-trained to align generated content with text semantics. The Image CA stream then performs attribute-level correspondence matching between the reference and noisy image features. It leverages the inherent ability of cross-attention to model feature similarity and dependencies (Vaswani et al., 2017; Ye et al., 2023): calculating attention scores between reference image tokens (encoded as K/V) and prompt-conditioned noisy image tokens (encoded as Q). Here, the noisy image serves as a bridge to connect the text prompt and reference image—only those reference attributes that align with the semantic priorities defined by Text CA are retained (with high attention scores), while conflicting attributes are suppressed (with low attention scores). The dual CA streams form a closed-loop decision-making process: Text CA defines "what to generate" (prompt alignment), while Image CA specifies "how to borrow from the reference" (attribute matching), thereby algorithmically determining whether to adopt reference attributes (when they match prompt priorities) or replace them (when they conflict with prompt priorities).

To intuitively illustrate this mechanism, we refer to the example in Fig. 16-Right-Row 2, where Van Gogh's Starry Night is used as the reference image:

- When the prompt is "A photograph of an old wooden bridge over a tranquil pond, rendered in **Impressionist swirls**": Text CA prioritizes low-level textural attributes ("Impression-

ist swirls") in the noisy image. In turn, Image CA assigns high attention scores to the prominent swirling textures in the reference image, enabling the targeted transfer of these low-level textural features.

- When the prompt is modified to "A photograph of an old wooden bridge over a tranquil pond, rendered in **melancholy tones**": Text CA now prioritizes the high-level semantic attribute ("melancholy tones") in the noisy image. Correspondingly, Image CA shifts to allocate higher attention scores to the reference's color palette and mood (high-level semantic attributes), resulting in the transfer of these abstract, high-level elements rather than the low-level swirling textures.

However, as analyzed in Sec. B.6 and further evidenced by the results in the dashed boxes of Fig. 16-Right, the retention of concept-agnostic information of the reference image may lead to subject copy-paste artifacts and compromise the alignment between generated results and text prompts—particularly regarding modifications to style, appearance, or posture. By leveraging our proposed DDS, such concept-agnostic information is effectively disentangled and eliminated, thereby further reinforcing the model's prompt-following competence and semantic disentanglement precision. Note that our DDS does not directly decide which elements to retain or modify but strengthens DCA's semantic focus by eliminating concept-agnostic noise from the reference image.

Currently, the feature extraction and fusion stage operates independently of the image generation stage, implying that adjustments to text prompts do not affect the expert fusion coefficients predicted by HMoE-FFM. The guidance for determining which elements (*e.g.*, content, style, high-level semantic attributes, or low-level textural details) of the reference image to retain or modify in the generated output is automatically achieved by DCA and further enhanced by our DDS. A promising future research direction is to establish a connection between the feature extraction/fusion stage and the image generation stage, enabling the prediction of expert fusion coefficients based on both input reference images and text prompts. We plan to explore this direction in our future work.

### B.8 MORE ABLATION STUDIES ON EXPERTS' LAYERS

Here, we present additional ablation results exploring alternative layer selections for the experts in HMoE-FFM, with a specific focus on the low-level and mid-level experts. As demonstrated in Fig. 17, we visualize the reconstruction performance of features from various shallow and middle layers by injecting them individually via cross-attentions. As shown in the top row, features from excessively shallow layers (e.g., layer 4) tend to introduce low-level artifacts, while near-middle layers (e.g., layer 12) fail to fully recover low-level details such as the text on the cloth. In contrast, layers 8 and 10 yield much more satisfactory results in terms of low-level detail preservation, with layer 10 producing slightly more natural visual outputs. As illustrated in the bottom row, features from near-shallow layers (e.g., layer 12) and near-deep layers (e.g., layer 21) are unable to restore fine-grained details like facial features. Middle layers 15 and 17, however, deliver significantly more satisfying results, with layer 17 generating marginally more natural outcomes. Therefore, we select features from layer 10 and layer 17 as the inputs for the low-level and mid-level expert networks, respectively.

## C LIMITATIONS AND DISCUSSIONS

The primary limitation of our DynaIP stems from its reliance on explicit mask-guided feature injection for achieving multi-subject personalization. Although this design enhances the practical flexibility of our method to a certain extent, mask extraction may give rise to issues owing to the inherent precision limitations of grounding and segmentation models (Liu et al., 2024; Kirillov et al., 2023). Furthermore, since our method leverages the generative prior of the base model, the performance of concept preservation may deteriorate if the subjects generated by the base model deviate significantly from the reference subjects, as illustrated in Fig. 18. Notably, this problem can be alleviated by employing prompts that align more closely with the subjects in the reference images.

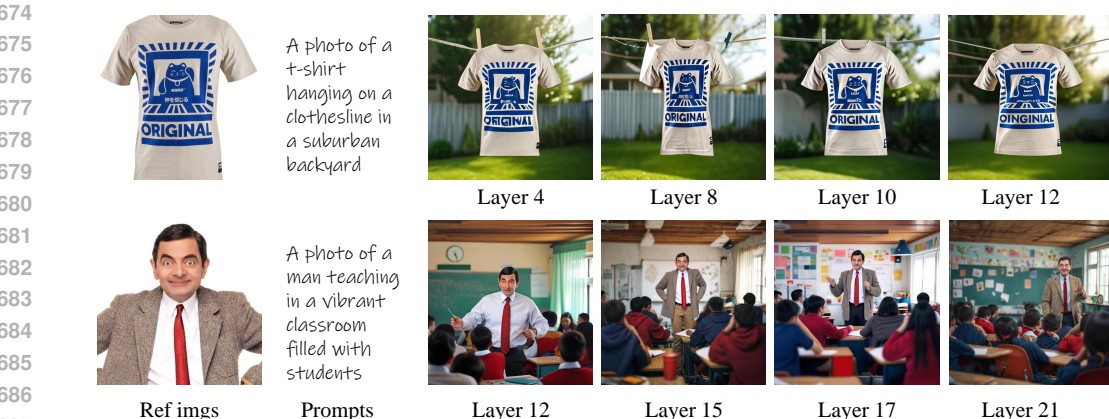

Figure 17: Personalization results generated by injecting features from **more layers of CLIP** via cross-attentions.

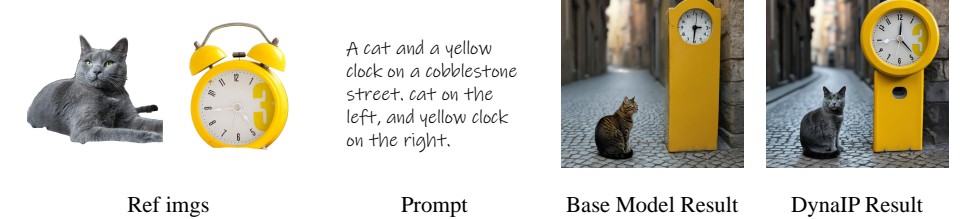

Figure 18: **Limitation of DynaIP**. The performance of concept preservation may degrade if the subjects generated by the base model deviate significantly from the reference subjects.

## D    ETHICS STATEMENT

As a finetuning-free personalized text-to-image generation method, DynaIP empowers users to flexibly create customized images for creative scenarios like digital design and storytelling, unlocking value for both professionals and casual users. Yet it also poses risks: its ability to generate realistic novel subject combinations may be misused to produce deceptive content, potentially fueling misinformation and eroding trust in visual media. Future efforts should focus on balancing creativity with risk mitigation—such as integrating content authentication tools and establishing ethical use guidelines—to ensure its positive societal impact.

## E    COPYRIGHT STATEMENT FOR IMAGES

Most images in this paper are sourced from open-access academic datasets and published literature. Their use strictly adheres to the terms of applicable open-source licenses or academic reproduction norms, with full attribution provided in the text and references. A small number of images were retrieved from public online sources. Despite reasonable efforts to verify their copyright status, such status remains unconfirmed. These images are used exclusively for non-commercial academic research purposes in this work.

## F    THE USE OF LARGE LANGUAGE MODELS

In this paper, we only used the Large Language Models (LLMs) to assist with text polishing. The LLMs played no role in the conception, design, or execution of the research.

