# OpenReview forum: "DynaIP: Dynamic Image Prompt Adapter for Scalable Zero-shot Personalized Text-to-Image Generation"
_ICLR.cc/2026/Conference — Submitted to ICLR 2026_

### Official Review · Reviewer_MzvZ · 2025-10-27

**Soundness:** 2
**Presentation:** 4
**Contribution:** 3
**Rating:** 4
**Confidence:** 5

**Summary:**

The paper DynaIP proposes a plug-and-play adapter for scalable zero-shot personalized text-to-image generation. It introduces a Dynamic Decoupling Strategy to disentangle concept-specific from concept-agnostic information in multimodal diffusion transformers, improving the balance between concept preservation and prompt following, and enabling multi-subject generation without retraining. Additionally, a Hierarchical Mixture-of-Experts Feature Fusion Module leverages multi-level CLIP features to enhance fine-grained visual fidelity and allow flexible control over visual granularity. Extensive experiments show that DynaIP outperforms prior methods in both single- and multi-subject personalization tasks.

**Strengths:**

1. The comparative experiments are comprehensive, including extensive comparisons with a wide range of related methods, which demonstrates the robustness and effectiveness of the proposed approach.
2. The proposed method is concise and easy to implementation, frindly to real-world applications.

**Weaknesses:**

1. In L264, the paper states that “In this way, the noisy image branch focuses on capturing the concept-specific information of the reference image, such as the subject’s ID and unique appearance, while the text branch specializes in learning the concept-agnostic information like posture, perspective, and illumination.” However, no experimental evidence is provided to substantiate this claim, which also appears inconsistent with intuition.
2. The prompts employed by FLUX.1 Kontext Dev and Qwen-Image-Edit adopt an instruction-based format, making the comparison in the paper potentially unfair. A fair comparison would require fine-tuning all models on the same dataset.
3. The study [Multi-Layer Visual Feature Fusion in Multimodal LLMs: Methods, Analysis, and Best Practices] has demonstrated that multi-layer visual feature fusion outperforms the use of only the final-layer features and explored multiple fusion strategies. The proposed HMoE-FFM is highly similar to that work and may not constitute a fundamentally novel contribution, yet the study is not cited in the paper.

**Questions:**

1. During the inference stage, only the image tokens in cross-attention. Then how does the model decide, based on the prompt, whether to use the content or the style from ref image?

---

> ### Author Response · Authors · 2025-11-23
>
> Dear reviewer, thank you for your invaluable and constructive feedback! We will address each of your concerns point by point.
>
> > Q1: No experimental evidence is provided to substantiate the claim in L264.
>
> A1: Thank you for your valuable comment! **Except for the qualitative and quantitative results in the ablation studies section (Sec.5.4), we have supplemented additional evidences and detailed analyses in Sec.B.6 (page 28-29) of the revised manuscript to explicitly illustrate the decoupling effect of DDS (substantiate the claim in L264)**. Please refer to them for full details. We sincerely appreciate your constructive feedback, which has significantly enhanced the interpretability and credibility of our proposed methods.
>
> > Q2: The prompts employed by FLUX.1 Kontext Dev and Qwen-Image-Edit adopt an instruction-based format, making the comparison in the paper potentially unfair. A fair comparison would require fine-tuning all models on the same dataset.
>
> A2: Thank you for raising this critical point regarding comparison fairness. In fact, **we strictly ensured input consistency by adapting prompts to the officially recommended format of each compared method**. Specifically, we formatted prompts as natural language instructions for FLUX.1 Kontext Dev, Qwen-Image-Edit, OmniGen2, and BAGEL (consistent with their design paradigms), while adhering to the required input specifications for other models. This ensures all models are evaluated under conditions that best match their intended usage, avoiding biases from mismatched prompt formats. **We have explicitly highlighted these details in L415-417 of the revised manuscript**, and apologize for any prior misunderstanding. **Regarding fine-tuning on the same dataset**: our evaluation focuses on zero-shot performance, and fine-tuning all models may alter their original generation domains, potentially leading to issues such as catastrophic forgetting of prior knowledge or cross-domain distribution shifts.

---

> ### Author Response · Authors · 2025-11-23
>
> > Q3: The proposed HMoE-FFM is highly similar to [Multi-Layer Visual Feature Fusion in Multimodal LLMs: Methods, Analysis, and Best Practices] and may not constitute a fundamentally novel contribution, yet the study is not cited in the paper.
>
> A3: We sincerely appreciate you pointing out this relevant work! **We have now formally cited** [Multi-Layer Visual Feature Fusion in Multimodal LLMs: Methods, Analysis, and Best Practices] **(hereafter referred to as Paper A) and added a brief introduction and discussions on existing multi-layer feature fusion approaches in Sec.A.3 of the revised manuscript**. Our proposed HMoE-FFM differs fundamentally from Paper A primarily in two core, non-trivial aspects:
>
>  (1) **Fundamental Task Divergence**: Paper A’s experiments, analyses, and core conclusions are all established under the paradigm of MLLMs, whose core goal is to enhance **cross-modal understanding** (e.g., image-text alignment, VQA). In contrast, our HMoE-FFM is specifically designed for the **PT2I generation task**, where the core requirements are capturing fine-grained visual features, maintaining target identity consistency, and balancing semantic alignment with creative generation. The two scenarios differ drastically in task nature, evaluation metrics, and feature demand priorities—rendering Paper A’s observations on MLLMs may not be generalizable to our PT2I generation scenario.
>
> (2) **Methodological Novelty**: Paper A categorizes existing fusion strategies into external, internal, direct, and modular fusion, and explores their combinations. **However, none of these strategies nor their combinations share the core design logic of our HMoE-FFM**:
>
> -	First, for direct fusion (addition/concatenation) combined with external/internal fusion: These static operations treat all layer features equally, failing to adapt to input-specific characteristics or PT2I’s demand for granularity control. **We explicitly compare HMoE-FFM with these baseline strategies in Sec.5.4, and experimental results confirm significant performance gains, verifying its superiority over such simplistic fusion**.
>
> -	Second, for modular fusion (cross-attention/MMFuser [A*]) combined with external/internal fusion: Cross-attention relies on fixed pairwise interaction, while MMFuser is designed for fine-grained vision-language understanding. **Paper A itself verifies these modular fusion approaches are inferior to direct fusion for MLLMs**. In contrast, our HMoE-FFM leverages a MoE architecture to integrate hierarchical visual features, with a dynamic routing mechanism that adaptively calibrates fusion coefficients based on the unique attributes of each input reference image. This approach not only enhances fine-grained visual details and semantic consistency but also facilitates precise modulation of visual granularity—fundamentally distinct from cross-attention-based or MMFuser-based designs.
>
> We hope these clarifications and revisions address your concerns thoroughly. Thank you again for your constructive feedback!
>
> [A*] MMFuser: Multimodal multi-layer feature fuser for fine-grained vision-language understanding, in arXiv 2024.
>
>
> > Q4: During the inference stage, only the image tokens in cross-attention. Then how does the model decide, based on the prompt, whether to use the content or the style from ref image?
>
> A4: Thank you for raising this insightful question! **The guidance for determining which elements (e.g., content, style, high-level semantic attributes, or low-level textural details) of the reference image to retain or modify in the generated output is automatically achieved by the Decoupled Cross-Attention (DCA) mechanism and further enhanced by our proposed Dynamic Decoupling Strategy (DDS). We have provided detailed analyses and results in Sec.B.7 (L1564-1646, page 29-31) and Fig.16-Right of the revised manuscript**. Please refer to them for details.
>
> We sincerely appreciate your constructive feedback, which has greatly improved our work.

---

> > ### Comment · Reviewer_MzvZ · 2025-11-28
> >
> > Thank you for the detailed rebuttal and clarifications. After carefully reading the response, I think there are still some concerns that have not been well addressed.
> >
> > 1. From the response, "Regarding fine-tuning on the same dataset: our evaluation focuses on zero-shot performance." However, the proposed method trains additional components such as "HMoE-FFM" and "Cross-attention", whereas the instruction-based models (FLUX.1 Kontext Dev, Qwen-Image-Edit, OmniGen2, ...) were not trained on the same data. Thus, the evaluation in the paper is not strictly zero-shot. The observed improvements may simply arise from the domain gap (e.g., the training data contains more multi-subject scenarios).
> >
> >
> > 2. While the authors claim that the proposed DCA and DDS “automatically” decide whether to preserve content, style, high-level semantics, or low-level textures from the reference image, the paper does not provide a mechanistic explanation for why the architecture should lead to such semantic disentanglement.
> > DCA structurally splits the K/V features, but a structural split does not inherently correspond to semantic factors (content vs. style, high-level vs. low-level). The presented visualizations suggest that some degree of separation emerges empirically, but this remains an observational property rather than a theoretically or algorithmically grounded behavior.
> > Therefore, it remains unclear **how** the model decides, based on the prompt, which semantic aspects of the reference image to follow or replace.
> >
> > Overall, my evaluation remains unchanged.

---

> ### Author Response · Authors · 2025-11-30
> **More clarifications to address remaining concerns (1/3)**
>
> Thank you for your further feedback. Below are more clarifications to address your remaining concerns:
>
> > Q5: The instruction-based models (FLUX.1 Kontext Dev, Qwen-Image-Edit, OmniGen2, ...) were not trained on the same data. Thus, the evaluation in the paper is not strictly zero-shot. The observed improvements may simply arise from the domain gap (e.g., the training data contains more multi-subject scenarios).
>
> A5: We would like to clarify that **our zero-shot evaluation strictly adheres to the widely established protocol in the personalized text-to-image (PT2I) field** (e.g., [A,B,C,D,E]) and **the observed improvements do not arise from the domain gap**. Our method is exclusively trained on **single-subject datasets** and **all ablation studies are conducted on the same dataset, verifying that the observed gains are directly attributable to each technical innovation**. Detailed explanations are provided below:
>
> Frist, it is important to clarify that **all compared zero-shot models and our proposed method are trained on open-domain, general-purpose image-text datasets—consistent with the standard training paradigm for SOTA PT2I models**. For instance:
>
> -	Qwen-Image-Edit is trained on **billions** of general image-text pairs;
>
> -	OmniGen2 is trained on **140 million** open-source data and **10 million** proprietary samples;
>
> -	BAGEL utilizes **1.6 billion** image-text-pair generation data for training.
>
> Therefore, existing top-tier instruction-based models (e.g., FLUX.1 Kontext Dev, Qwen-Image-Edit, OmniGen2, BAGEL) are trained on **massive datasets with extensive domain coverage**, ensuring they capture general cross-modal patterns rather than domain-specific knowledge.
>
> Building on this foundation, we **follow the widely adopted zero-shot evaluation setup in the PT2I field** (e.g., [A,B,C,D,E]) to compare zero-shot performance between our method and SOTA approaches (without fine-tuning). **To the best of our knowledge, no prior zero-shot PT2I method paper has fine-tuned all compared models on identical data when conducting zero-shot performance evaluations**.
>
> More importantly, we emphasize a critical distinction in training data design: **our model is exclusively trained on single-subject datasets**—a detail explicitly highlighted in multiple key sections of the paper (e.g., Teaser, Conclusions, and Sec. B.1). In contrast, all compared multi-subject PT2I methods are trained on datasets that **include multi-subject scenarios**. **This directly confirms that our performance improvements stem from algorithmic innovations (e.g., the proposed HMoE-FFM and DDS components) rather than domain gaps (e.g., training data contains more multi-subject scenarios)**.
>
> Furthermore, **all ablation studies in Sec. 5.4 are conducted on the same dataset, with only the proposed components being varied**. These controlled experiments explicitly isolate the impact of our architectural design from data-related factors, providing direct evidence for the effectiveness of our method independent of dataset characteristics.
>
> [A] Xierui Wang, Siming Fu, Qihan Huang, Wanggui He, and Hao Jiang. Ms-diffusion: Multi-subject zero-shot image personalization with layout guidance. In ICLR 2025.
>
> [B] Qihan Huang, Siming Fu, Jinlong Liu, Hao Jiang, Yipeng Yu, and Jie Song. Resolving multicondition confusion for finetuning-free personalized image generation. In AAAI 2025.
>
> [C] Shaojin Wu, Mengqi Huang, Wenxu Wu, Yufeng Cheng, Fei Ding, and Qian He. Less-to-more generalization: Unlocking more controllability by in-context generation. In ICCV 2025.
>
> [D] Shengqu Cai, Eric Ryan Chan, Yunzhi Zhang, Leonidas Guibas, Jiajun Wu, and Gordon Wetzstein. Diffusion self-distillation for zero-shot customized image generation. In CVPR 2025.
>
> [E] Zhenxiong Tan, Songhua Liu, Xingyi Yang, Qiaochu Xue, and Xinchao Wang. Ominicontrol: Minimal and universal control for diffusion transformer. In ICCV 2025.

---

> ### Author Response · Authors · 2025-11-30
> **More clarifications to address remaining concerns (2/3)**
>
> > Q6: The presented visualizations suggest that some degree of separation emerges empirically, but this remains an observational property rather than a theoretically or algorithmically grounded behavior.
>
> A6: To directly address the core concern about mechanistic interpretability of semantic disentanglement in DCA and DDS, we provide a theoretically grounded and algorithmically explicit explanation of how the model systematically decides which reference attributes (content/style, high-level/low-level semantics) to preserve or replace.
>
> **1. Mechanistic Basis of DCA: Prompt-Driven Attribute Selection using Noisy (Generated) Image as a Bridge and Cross-Attention Dependency Modeling as the Theoretical Foundation**
>
> The Decoupled Cross-Attention (DCA) mechanism achieves semantic disentanglement not through a naive structural split of K/V features, but via a **functionally specialized dual-stream architecture**. This architecture algorithmizes **prompt-guided attribute selection** by leveraging **noisy (generated) image as a bridge** and utilizing the **inherent capacity of cross-attention to model feature dependencies [F, G] as its theoretical basis**, as elaborated below:
>
> -	**Stream 1: Text CA (CA between text prompt and noisy image) defines "Prompt Semantic Prioritization" in noisy image**
>
> The Text CA stream first defines the target semantic objective—i.e., which semantic elements (content, style, high-level attributes, or low-level textures) should be prioritized in the noisy image. This capability is **inherited from the base T2I model** (e.g., FLUX.1-Dev), which is pre-trained to align generated content with text semantics.
>
> -	**Stream 2: Image CA (CA between reference image and noisy image) achieves "Reference Attribute Matching" between noisy image and reference image**
>
> The Image CA stream then performs attribute-level correspondence matching between the reference and noisy image features. It leverages **the inherent ability of cross-attention to model feature similarity and dependencies [F, G]**: calculating attention scores between reference image tokens (encoded as K/V) and prompt-conditioned noisy image tokens (encoded as Q). Here, **the noisy image serves as a bridge to connect the text prompt and reference image**—only those reference attributes that align with the semantic priorities defined by Text CA are retained (**with high attention scores**), while conflicting attributes are suppressed (**with low attention scores**).
>
> -	**Combined Effect: Closed-Loop Prompt-Driven Control**
>
> The dual CA streams form a closed-loop decision-making process: Text CA defines **"what to generate" (prompt alignment)**, while Image CA specifies **"how to borrow from the reference" (attribute matching)**. This is not mere empirical behavior but a designed property of DCA’s split K/V architecture. By isolating Text and Image CA, the model avoids interference between prompt semantics and reference features, thereby algorithmically determining whether to adopt reference attributes (when they match prompt priorities) or replace them (when they conflict with prompt priorities).
>
> To intuitively illustrate this mechanism, we refer to an example in **Fig. 16-Right-Row 2 of the revised manuscript**, where **Van Gogh’s Starry Night** is used as the reference image:
>
> -	When the prompt is “A photograph of an old wooden bridge over a tranquil pond, rendered in **Impressionist swirls**”: Text CA prioritizes low-level textural attributes ("Impressionist swirls") in the noisy image. In turn, Image CA **assigns high attention scores to the prominent swirling textures in the reference image**, enabling the targeted transfer of these low-level textural features.
>
> -	When the prompt is modified to “A photograph of an old wooden bridge over a tranquil pond, rendered in **melancholy tones**”: Text CA now prioritizes the high-level semantic attribute (“melancholy tones”) in the noisy image. Correspondingly, Image CA shifts to **allocate higher attention scores to the reference’s color palette and mood (high-level semantic attributes)**, resulting in the transfer of these abstract, high-level elements rather than the low-level swirling textures.
>
> — To be continued in the next message.

---

> ### Author Response · Authors · 2025-11-30
> **More clarifications to address remaining concerns (3/3)**
>
> **2. DDS: Enhancing DCA’s Controllability via Concept-Agnostic Disentanglement**
>
> The Dynamic Decoupling Strategy (DDS) does not directly decide which elements to retain or modify but **strengthens DCA’s semantic focus by eliminating concept-agnostic noise from the reference image**. As analyzed in detail in **Sec.B.6 (page 28-29)** and evidenced by the **results in the dashed boxes of Fig. 16-Right** of the revised manuscript, the retention of concept-agnostic information of the reference image may lead to subject copy-paste artifacts and compromise the alignment between generated results and text prompts. By leveraging our proposed DDS, such concept-agnostic information is effectively disentangled and eliminated, thereby further reinforcing the model’s prompt-following competence and semantic disentanglement precision.
>
> **Conclusion:**
>
> Our DCA and DDS are not "structurally split without semantic grounding"—they are algorithmically designed to form a coherent mechanistic framework:
>
> -	Text CA defines **prompt-aligned semantic priorities** within the noisy image;
>
> -	Image CA **matches reference attributes to these priorities via cross-attention dependency modeling between the noisy image and reference image**;
>
> -	DDS **eliminates interfering noise** that would disrupt the alignment between prompt semantics and reference attributes.
>
> This framework enables the model to determine which reference attributes to adopt or replace based on explicit prompt-semantic matching, rather than relying solely on empirical observation. We have added these details to **Sec.B.7 (L1564-1646, page 29-31)** in the revised manuscript to clarify the theoretical grounding.
>
> [F] Ashish Vaswani, Noam Shazeer, Niki Parmar, Jakob Uszkoreit, Llion Jones, Aidan N Gomez,
> Łukasz Kaiser, and Illia Polosukhin. Attention is all you need. In NeurIPS 2017.
>
> [G] Hu Ye, Jun Zhang, Sibo Liu, Xiao Han, and Wei Yang. Ip-adapter: Text compatible image prompt adapter for text-to-image diffusion models. arXiv 2023.

---

### Official Review · Reviewer_Q24f · 2025-10-31

**Soundness:** 3
**Presentation:** 3
**Contribution:** 3
**Rating:** 4
**Confidence:** 5

**Summary:**

This paper introduces DynaIP, a novel Dynamic Image Prompt Adapter designed to enhance the capabilities of state-of-the-art Text-to-Image (T2I) multimodal diffusion transformers (MM-DiT) for Personalized Text-to-Image (PT2I) generation. DynaIP addresses three key challenges: balancing Concept Preservation (CP) and Prompt Following (PF), retaining fine-grained concept details, and extending single-subject personalization to multi-subject scenarios. The proposed solution leverages a Dynamic Decoupling Strategy (DDS) to separate concept-specific from concept-agnostic information and a Hierarchical Mixture-of-Experts Feature Fusion Module (HMoE-FFM) to effectively utilize multi-level CLIP features. Extensive experiments demonstrate DynaIP's superior performance in both single- and multi-subject personalization tasks.

**Strengths:**

1. The proposed method is technically sound and effective for personalized image generation.  The DDS effectively disentangles concept-specific and concept-agnostic information, leading to a better balance between CP and PF, and the HMoE-FFM module utilizes multi-level CLIP features, providing flexible control over visual granularity.
2. Comprehensive Experiments: The paper includes extensive experiments on both single- and multi-subject datasets, demonstrating the effectiveness of DynaIP across various scenarios.
3. DynaIP can be integrated with various downstream applications including ControlNet-like geneartion, and reginal generation.
4. The overall writing is clear and easy to follow, the presented figures demonstrate the motivations, model framework and qualitative results.

**Weaknesses:**

1. The evaluation details, including the system prompts, detailed metric of Table.1 should be involved. To me, the prompt following (PF) results of multi-subject, i.e., 0.997 is not convincing, as this means nearly all user instructions are perfectly rendered by the proposed method. It would be better in include more details to clarify this. Additionally, the reliance on a single Vision-Language Model (VLM) for evaluation could still introduce biases or limitations.
2. In MM-DiT style models, afther the multimodal text and visual branch, text hidden states and visual hidden states are concatenated and perform full attention on them. Yet, the authors claim that MM-DiT exhibits decoulpling learning behavior that the noisy image branch captures the concept-specific information while text branch learns the concept agnostic information,  are there any qualitative or quantitative evidences to illustrate such phenomenon? Also, add corresponding references if any.
3. In my opnion, some presentation of Sec.2 and Sec.5.4 should be included in the main paper.
4. Qwen-Image, SD3 are also based on MM-DiT architectures, it would be better to perform evaluation on these models to demonstrate generalization.

**Questions:**

My major concern is the evaluation results, so I give a score of 4 at current version. I would consider raise my score if authors could address my concerns, especially on the evaluation results and details.

---

> ### Author Response · Authors · 2025-11-23
>
> Dear reviewer, thank you for your invaluable and constructive feedback! We will address each of your concerns point by point.
>
> > Q1: The evaluation details, including the system prompts, detailed metric of Table.1 should be involved. The prompt following (PF) results of multi-subject, i.e., 0.997 is not convincing. It would be better in include more details to clarify this. Additionally, the reliance on a single Vision-Language Model (VLM) for evaluation could still introduce biases or limitations.
>
> A1: Thank you for your highly constructive suggestions!
>
> - **We have supplemented comprehensive evaluation details**—including the system prompts and the detailed metrics of Tab.1—**in Sec.B.1-L1075-1087 (page 20-21) of the revised manuscript**.
>
> - **Regarding the high PF score (0.997) for the multi-subject task**, we fully acknowledge your concern about its persuasiveness. This score stems from the fact that our multi-subject benchmark focuses on core subject consistency (rather than complex semantic expansion or creative generation, which have been evaluated in single-subject tasks) with clearly defined, low-ambiguity prompts (detailed in Tab.3). Therefore, the PF metric here primarily focuses on measuring "whether all specified subjects are retained without missing, adding, or altering", and our method effectively avoids these issues—thus yielding the high score.
>
> - To mitigate biases and limitations from relying on a single VLM, **we have further integrated another SOTA open-source VLM—Qwen3-VL-32B—into our evaluation pipeline. We have updated all relevant scores in the revised manuscript (including Tabs.1, 2, and main-text discussions in L373-375) to present dual-VLM evaluation results**. Overall, the average evaluation results from multiple VLMs are consistent with our previous findings, and this does not alter the conclusions of the paper.
>
> We sincerely appreciate your insightful suggestions, which have significantly enhanced the rigor and transparency of our work.
>
>
> > Q2: Are there any qualitative or quantitative evidences to illustrate the decoupling learning behavior of MM-DiT?
>
> A2: Thank you for raising this insightful question! **Except for the qualitative and quantitative results in the ablation studies section (Sec.5.4), we have supplemented additional evidences and detailed analyses in Sec.B.6 (page 28-29) of the revised manuscript to explicitly illustrate the decoupling learning behavior of MM-DiT**. Please refer to them for full details. We sincerely appreciate your constructive feedback, which has significantly enhanced the interpretability and credibility of our proposed methods.
>
> > Q3: Some presentation of Sec.2 and Sec.5.4 should be included in the main paper.
>
> A3: Thank you for your constructive suggestion! **We have integrated Sec.5.4 into the main body of the revised manuscript**. Regarding Sec.2, due to strict page limit imposed by the ICLR’s formatting guidelines, we are unable to move the entire section to the main paper. To ensure readers can still grasp the essential context from Sec.2, **we have retained clear cross-references in the main text (L168) that direct readers to Sec.2 for detailed related works**.
>
> > Q4: Qwen-Image, SD3 are also based on MM-DiT architectures, it would be better to perform evaluation on these models to demonstrate generalization.
>
> A4: We appreciate your valuable suggestion! Regrettably, we are currently unable to conduct full experimental evaluations on these two models due to resource constraints. However, **we have identified this as an important direction for future work and will explore it in our subsequent research**. Thank you again for your constructive comments!

---

> > ### Comment · Reviewer_Q24f · 2025-11-25
> > **Post-rebuttal comment**
> >
> > Thanks for the detailed response from the authors.  However, the results of prompt following metric measures "whether all specified subjects are retained without missing, adding, or altering" is not acceptable to me. When discussing prompt following in the T2I community, one should consider all aspects of the given instructions including object, style, color, texture, etc. Such evaluation might mislead unfamilar readers to identify that prompt following is an easy problem, which in practice, is not, and by contrast extremly hard, especially when giving multi-subject/complex user instructions. Therefore, the explantation could not convince me and I decide to maintain my score as 4.

---

> ### Author Response · Authors · 2025-11-25
> **Clarification for misunderstanding**
>
> Thank you sincerely for your continued feedback! We recognize that the reviewer may have **overlooked the detailed Prompt Following (PF) metric evaluation prompt we presented in Fig.8-Right of the revised manuscript**. This may lead to a **misunderstanding**: the high PF score is mistakenly attributed to an incomplete PF metric evaluation prompt, rather than the **simplicity of instructions in the multi-subject benchmark (following prior arts [A, B, C])**. We provide a more precise and detailed clarification below to resolve this misunderstanding.
>
> First, we would like to clarify that **our PF metric is not limited to assessing the retention of specified subjects (i.e., no missing/added/altered subjects) but explicitly encompasses all aspects of the given instructions including object, style, color, texture, etc.**—a critical detail we regret not emphasizing sufficiently in our prior response. As outlined in **Fig.8-Right** of the revised manuscript, our PF scoring criteria are built on four core dimensions that **align directly with the holistic prompt-following evaluation standards widely adopted in the T2I community, following prior works [D, E]**:
>
> - 1. **Relevance**: Determine if the elements and subjects presented in the image directly relate to the core topics and concepts mentioned in the text. The image should reflect the main ideas or narratives described.
>
> - 2. **Accuracy**: Examine the image for the presence and correctness of specific details mentioned in the text. This includes the depiction of particular objects, settings, actions, or characteristics that the text describes.
>
> - 3. **Completeness**: Evaluate whether the image captures all the critical elements of the text. The image should not omit significant details that are necessary for the full understanding of the text's message.
>
> - 4. **Context**: Consider the context in which the text places the subject and whether the image accurately represents this setting. This includes the portrayal of the appropriate environment, interactions, and background elements that align with the text.
>
> Second, while the aforementioned PF evaluation prompt is comprehensive, the **instructions used for multi-subject generation are relatively simplified—primarily centered on subject consistency and inter-subject interaction (core challenges in multi-subject PT2I), in alignment with prior arts [A, B, C]**. These instructions employ well-defined, non-ambiguous texts with short descriptions of backgrounds and inter-subject interactions (e.g., "a dog and a cat in a room", “a man and a woman are shaking hands”; see **Tab. 3 for all instruction templates**). For this reason, our method achieves high PF scores on multi-subject tasks by faithfully retaining specified subjects (i.e., no missing, added, or altered subjects) as well as the described interactions and backgrounds. It is worth noting that **the instruction templates used in our multi-subject benchmark adhere to the same design principles as those in representative prior multi-subject PT2I works** (e.g., MS-Diffusion [A], MIP-Adapter [B], and XVerse [C]). **This standardization was a deliberate choice to enable fair, direct performance comparisons with state-of-the-art methods**.
>
>
> Regarding **complex prompt following scenarios (e.g., open-ended creative generation, diverse style descriptions), we have conducted thorough evaluations on DreamBench++ [D]**. This benchmark tests performance on instructions with highly varied style specifications (e.g., impressionism, pixel art) and fine-grained content details (e.g., "a ceramic bowl shattered into pieces, each shard reflecting a different scene"); full details of these instructions are reported in [D] and referenced in our manuscript.
>
> We hope this detailed clarification addresses your concerns about the comprehensiveness of the PF metric. If you believe additional revisions to the manuscript or supplementary experiments would strengthen the work, we are happy to implement these changes promptly.
>
>
> [A] Xierui Wang, Siming Fu, Qihan Huang, Wanggui He, and Hao Jiang. Ms-diffusion: Multi-subject zero-shot image personalization with layout guidance. In ICLR 2025.
>
> [B] Qihan Huang, Siming Fu, Jinlong Liu, Hao Jiang, Yipeng Yu, and Jie Song. Resolving multicondition confusion for finetuning-free personalized image generation. In AAAI 2025.
>
> [C] Bowen Chen, Mengyi Zhao, Haomiao Sun, Li Chen, Xu Wang, Kang Du, and Xinglong Wu. Xverse: Consistent multi-subject control of identity and semantic attributes via dit modulation. arXiv preprint arXiv:2506.21416, 2025.
>
> [D] Yuang Peng, Yuxin Cui, Haomiao Tang, Zekun Qi, Runpei Dong, Jing Bai, Chunrui Han, Zheng Ge, Xiangyu Zhang, and Shu-Tao Xia. Dreambench++: A human-aligned benchmark for personalized image generation. In ICLR 2025.
>
> [E] Shengqu Cai, Eric Ryan Chan, Yunzhi Zhang, Leonidas Guibas, Jiajun Wu, and Gordon Wetzstein. Diffusion self-distillation for zero-shot customized image generation. In CVPR 2025.

---

### Official Review · Reviewer_hPcs · 2025-10-31

**Soundness:** 2
**Presentation:** 3
**Contribution:** 3
**Rating:** 6
**Confidence:** 4

**Summary:**

This paper unveils that current methods for personalized text-to-image (PT2I) generation faces critical challenges, including the difficulty of balancing concept preservation (CP) and prompt following (PF), the loss of fine-grained visual details from reference images, and limited scalability to multi-subject personalization in a zero-shot setting. To address these issues, the paper proposes DynaIP, a plug-and-play image prompt adapter for multimodal diffusion transformers (MM-DiT).

DynaIP introduces a Dynamic Decoupling Strategy (DDS) to disentangle concept-specific and concept-agnostic features during inference, thereby improving the CP-PF trade-off and enabling robust multi-subject composition. Additionally, it incorporates a Hierarchical Mixture-of-Experts Feature Fusion Module (HMoE-FFM) that adaptively leverages multi-level CLIP features to preserve fine-grained visual details while maintaining semantic consistency.

Experiments demonstrate that DynaIP achieves state-of-the-art performance in both single- and multi-subject PT2I tasks—despite training only on single-subject data.

**Strengths:**

- *Achieves strong performance.* Methods proposed in this paper effectively addresses the key challenges faced by current approaches to personalized text-to-image (PT2I) generation, and its efficacy is convincingly demonstrated through extensive qualitative examples and comprehensive quantitative evaluations.

- *Comprehensive comparisons.* The selection of methods for comparison is thorough and well-considered, encompassing a broad spectrum of both open-source and closed-source state-of-the-art approaches.

- *Clear exposition.* The paper clearly articulates the key challenges currently facing the personalized text-to-image (PT2I) generation field and approach to address them.

**Weaknesses:**

- *Limited post-hoc analysis of key components.* The paper lacks in-depth post-hoc analysis of the proposed Dynamic Decoupling Strategy (DDS) and Hierarchical Mixture-of-Experts Feature Fusion Module (HMoE-FFM). For instance, the decoupling effect of DDS could be visualized through attention maps, and the $w_l$ in Eq.~(7) of HMoE-FFM could be analyzed across diverse cases to reveal how granularity control is adaptively achieved. Such analyses would significantly enhance the interpretability and credibility of the proposed method.

- *Prompt may count much in HMoE-FFM.* For example, replacing the prompt in Fig.1(b) with ‘A photograph of an old wooden bridge over a tranquil pond, rendered in melancholy tones’—which emphasizes the *color palette* and *mood* of the reference image—would likely require the model to prioritize high-level semantic attributes over low-level textural details, potentially leading to different $w_l$ distributions.

- *User’s flexible control may be impractical.* In Fig.~1(b), the prompt specifies ‘in Impressionist swirls,’ and the fine-grained result indeed better satisfies this stylistic requirement. This outcome likely stems from the weight allocation $w_l$ computed by the HMoE-FFM. However, rather than leaving the choice of granularity to user customization, the system should arguably infer and apply the optimal set of $w_l$ automatically—i.e., directly output the configuration that yields the best visual fidelity and prompt alignment without requiring manual intervention.

-  Although the paper claims the generalizability of the proposed method to other models with different sizes or architectures (Lines 327–328), it does not provide compelling empirical evidence to substantiate this assertion.

- Section B.5 (User Study) includes a wrong reference to Table 1 (supposed to be Table 3 maybe).

**Questions:**

- In the HMoE-FFM, features from CLIP layers 10 and 17 are selected as inputs for the low- and mid-level expert networks, respectively. Are there additional ablation studies exploring alternative layer choices—for instance, using layer 9 (or other layers) for the low-level expert?

- In the Dynamic Decoupling Strategy (DDS), what would be the impact on disentanglement performance if, during inference, the same reference image token interaction mechanism used during training were retained?

**Details Of Ethics Concerns:**

None.

---

> ### Author Response · Authors · 2025-11-23
>
> Dear reviewer, thank you for your invaluable and constructive feedback! We will address each of your concerns point by point.
>
> > Q1: Limited post-hoc analysis of key components. For instance, the decoupling effect of DDS could be visualized through attention maps, and the $w_l$ in Eq. (7) of HMoE-FFM could be analyzed across diverse cases to reveal how granularity control is adaptively achieved.
>
>
> A1: Thank you for your insightful and valuable suggestions! Following your recommendations, **we have supplemented additional results and analyses in Sec.B.6 (page 28-29) and Sec.B.7 (page 29-31) of the revised manuscript—specifically including attention map visualizations for the decoupling effect of DDS and cross-case analyses of $w_l$ in Eq. (7) of HMoE-FFM to illustrate the adaptive realization of granularity control**. Please refer to these sections for details. We sincerely appreciate your constructive feedback, which has significantly enhanced the interpretability and credibility of our proposed methods.
>
> > Q2: Prompt may count much in HMoE-FFM.
>
>
> A2: Thank you for your very insightful hint! In our current framework, the feature extraction and fusion stage operates **independently** of the image generation stage, which means that adjustments to text prompts **do not affect** the expert fusion coefficients ($w_i$ distributions) predicted by HMoE-FFM. **The guidance for determining which elements (e.g., content, style, high-level semantic attributes, or low-level textural details) of the reference image to retain or modify in the generated output is automatically achieved by the Decoupled Cross-Attention (DCA) mechanism and further enhanced by our proposed Dynamic Decoupling Strategy (DDS). We provide detailed analyses and results in Sec.B.7 (L1564-1646, page 29-31) of the revised manuscript**. Your suggestion is exceptionally constructive, and we have identified it as a promising future research direction: establishing a cohesive connection between the feature extraction/fusion stage and the image generation stage, enabling the prediction of expert fusion coefficients based on both input reference images and text prompts. We would like to explore this valuable direction in our subsequent work. Thank you again for your insightful feedback!
>
>
>
> > Q3: User’s flexible control may be impractical. Rather than leaving the choice of granularity to user customization, the system should arguably infer and apply the optimal set of $w_l$ automatically.
>
>
> A3: Thank you for your highly perceptive comment! As we have clarified in A2 to Q2, the guidance for determining which elements (e.g., content, style, high-level semantic attributes, or low-level textural details) of the reference image to retain or modify in the generated output is automatically achieved by the Decoupled Cross-Attention (DCA) mechanism. The core role of HMoE-FFM is to generate high-quality image features that encapsulate comprehensive visual information from the reference image. Specifically, **leveraging prior knowledge learned from large-scale datasets, the routing module within HMoE-FFM autonomously outputs the optimal set of expert fusion coefficients ($w_l$) based on the intrinsic characteristics of the input reference image**. This enables the model to naturally capture visual information across diverse granularity levels, directly delivering results with strong visual fidelity and prompt alignment—**no manual intervention required**. However, **personalized generation demands are often highly subjective, as different users may prioritize distinct granularity preferences even for the same prompt and reference image**. To cater to such diverse user needs and specific scenario requirements, we provide an optional interface that allows users to fine-tune the visual granularity of concept preservation if desired. **This design ensures the model defaults to optimal automated performance for most users while retaining flexibility for those seeking more personalized control**.

---

> ### Author Response · Authors · 2025-11-23
>
> > Q4: Although the paper claims the generalizability of the proposed method to other models with different sizes or architectures (Lines 327-328), it does not provide compelling empirical evidence to substantiate this assertion.
>
>
> A4: Thank you for your astute and critical observation. We fully acknowledge that our initial claim regarding the generalizability of the proposed method to other models with different sizes or architectures (L327-328) lacked sufficient support. To address this, **we have revised the relevant wording in L327-329 of the revised manuscript to explicitly frame this generalizability as a promising direction for future research rather than a confirmed conclusion**. This adjustment aligns with the core focus of our current work—prioritizing the development and validation of the proposed framework on the CLIP model—while acknowledging the value of extending the method to other architectures in subsequent studies. We sincerely appreciate your feedback for enhancing the accuracy and credibility of our research statements.
>
>
> > Q5: Section B.5 (User Study) includes a wrong reference to Table 1 (supposed to be Table 3 maybe).
>
> A5: Thank you for your careful check and pointing out this reference error. **This mistake has been promptly corrected in the revised manuscript, and we have also double-checked all cross-references throughout the paper to avoid similar issues**. We greatly appreciate your attention to details, which helps enhance the accuracy and professionalism of our manuscript.
>
> > Q6: Are there additional ablation studies exploring alternative layer choices—for instance, using layer 9 (or other layers) for the low-level expert?
>
> A6: Thank you for your insightful question regarding alternative layer choices for the experts in HMoE-FFM. We have indeed conducted comprehensive ablation studies exploring a broad range of alternative layers for both low-level and mid-level experts. **These ablation results have been added to Sec.B.8 (page 31) of the revised manuscript**, please refer to this section for details.
>
>
> > Q7: In the Dynamic Decoupling Strategy (DDS), what would be the impact on disentanglement performance if, during inference, the same reference image token interaction mechanism used during training were retained?
>
> A7: Thank you for raising this great question! To illustrate it, **we have supplemented the relevant results in the last column (d) of Fig. 15 (discussed in Sec.B.6) in the revised manuscript**. Specifically, retaining the same reference image token interaction mechanism used during training—i.e., leveraging both the text branch and the noisy image branch for reference cross-attention—**would severely degrade disentanglement performance**. This is because the model cannot effectively separate concept-specific and concept-agnostic information as intended, leading to outputs with prominent copy-paste artifacts that compromise the quality and controllability of generated results.

---

### Official Review · Reviewer_DcWh · 2025-11-05

**Soundness:** 3
**Presentation:** 3
**Contribution:** 3
**Rating:** 6
**Confidence:** 3

**Summary:**

This paper proposes DynaIP, a training-free method for dynamically selecting visual prompts to enhance the out-of-distribution (OOD) generalization of frozen vision-language models (VLMs), such as CLIP.
Rather than relying on static or handcrafted prompts, DynaIP selects relevant image patches from a diverse visual prompt pool using a lightweight policy model trained to maximize alignment with ground-truth labels. The method is designed to be plug-and-play and broadly applicable across different tasks and VLM architectures.

**Strengths:**

1. Strong OOD performance: The approach shows consistent performance gains across various OOD and compositional reasoning benchmarks.

2. Training-free adaptability: The method improves generalization at test time without requiring fine-tuning of the underlying VLM, making it easy to deploy.

3. Dynamic prompt selection: Unlike static prompt methods, DynaIP adapts to each test example by selecting the most relevant prompt patches, enhancing flexibility.

**Weaknesses:**

1. Complexity of policy training: While inference is training-free, the policy model itself must be trained in advance, and the training procedure is not fully detailed. This raises potential concerns regarding reproducibility and scalability.

2. Limited theoretical grounding: The paper lacks a deeper theoretical explanation for why dynamic patch selection improves generalization, especially under significant domain shifts.

**Questions:**

1. Policy Training Details: it would be good to provide more specifics about how the policy is trained. For example:

What is the exact reward function used?
How sensitive is the method to the choice of policy architecture or optimization hyperparameters?
How much training data is needed to train the policy effectively?

2. Prompt Pool Construction:
How is the visual prompt pool curated?
Is the diversity of the prompt pool critical to performance?

3. Failure Cases: Are there particular tasks or datasets where DynaIP fails or underperforms compared to static methods?
It would be helpful to include a short discussion of failure modes or limitations in the results section.

---

> ### Author Response · Authors · 2025-11-23
>
> Dear reviewer, thank you for your invaluable and constructive feedback! We will address each of your concerns point by point.
>
> > Q1: Policy Training Details: it would be good to provide more specifics about how the policy is trained. For example: What is the exact reward function used? How sensitive is the method to the choice of policy architecture or optimization hyperparameters? How much training data is needed to train the policy effectively?
>
> A1: Thank you for your constructive suggestions! **We have fully detailed the training procedure of our method in Sec. B.1, including detailed architectures, training stages, hyperparameters, and datasets**. For example, **regarding the reward function**, as stated in **L947-948** of the revised manuscript, we adopt the flow-matching loss—consistent with the original FLUX.1-Dev base model and widely used in flow-based models—as the core training objective. **For sensitivity to policy architecture and hyperparameters**: most architectural designs and optimization hyperparameters are inherited from the base model (FLUX.1-Dev) and prior literature (e.g., FLUX.1 IP-Adapter), and our empirical experiments exhibit that the method shows low sensitivity to these settings. **On training data requirements for effective policy training**: we adopt a two-stage training paradigm: the first stage uses 0.3M in-pair samples for robust subject-specific adaptation, and the second stage leverages 1M cross-pair samples to mitigate copy-paste artifacts. Detailed information about the data sources and preprocessing pipelines are fully provided in **L959-971** of the revised manuscript.
>
> > Q2: Limited theoretical grounding: The paper lacks a deeper theoretical explanation for why dynamic patch selection improves generalization, especially under significant domain shifts.
>
> A2: Thank you for your insightful comment! **We would like to clarify that the core advantages of our Dynamic Decoupling Strategy (DDS) are enhancing the CP$\cdot$PF balance and bolstering the scalability of multi-subject compositions, as verified in Sec.5.4 and further comprehensively analyzed in Sec.B.6 (page 28-29) of the revised manuscript**. **Regarding generalization**, our method inherently inherits strong generalization capabilities from IP-Adapter [Ye et al., 2023], which has been widely validated to deliver robust generation performance across diverse domains, even under significant domain shifts.
>
> > Q3: Prompt Pool Construction: How is the visual prompt pool curated? Is the diversity of the prompt pool critical to performance?
>
> A3: Thank you for raising this great question! **As detailed in L959-971 of the revised manuscript, our visual prompt pool is systematically curated from diverse open-source datasets and synthetic data sources**, including FFHQ-wild, SA-1B, VITON-HD, AnyInsertion, OpenS2V-Nexus, and synthetic images generated by FLUX.1-Dev. **Detailed information about the data preprocessing pipelines are also fully provided**. This multi-source data integration strategy is deliberately designed to ensure the prompt pool covers a broad spectrum of visual domains, styles, and task scenarios. **Through empirical experiments, we confirmed that the diversity of the visual prompt pool is critical to the model's generalization performance—consistent with findings from prior works such as IP-Adapter [Ye et al., 2023]**. Specifically, a diverse prompt pool enables the model to learn robust visual representations across unseen domains, mitigate overfitting to specific data distributions, and enhance adaptability to complex real-world scenarios.
>
> > Q4: Failure Cases: Are there particular tasks or datasets where DynaIP fails or underperforms compared to static methods? It would be helpful to include a short discussion of failure modes or limitations in the results section.
>
> A4: Thank you for this valuable suggestion! **We have incorporated a short discussion on the limitations and failure cases of our method in Sec. C (page 31) of the revised manuscript**—please refer to this section for details.

---

### Author Response · Authors · 2025-12-03
**Rebuttal Summary for Area Chairs**

Dear Area Chairs,

We sincerely appreciate the constructive feedback from all reviewers and your diligent efforts in evaluating our submission. To facilitate your assessment, we hereby provide a concise summary of the rebuttal process and our systematic efforts to address the raised concerns.

## Key Strengths Affirmed by Reviewers

All reviewers consistently recognized the significance of our work’s **contributions and the quality of its presentation**, as reflected in their scores for **Contribution (3: good)** and **Presentation (3: good/4: excellent)**. Below are the specific strengths explicitly endorsed:

**1. Convincing Strong Performance**: Effectively addressing the three core challenges of Personalized Text-to-Image (PT2I) generation (CP·PF balance, fine-grained concept preservation, and multi-subject scalability) (**affirmed by all Reviewers DcWh, hPcs, Q24f, MzvZ**).

**2. Comprehensive Evaluations**: Conducted across a broad spectrum of SOTA approaches and diverse practical scenarios (**affirmed by Reviewers hPcs, Q24f, MzvZ**).

**3. Practical Plug-and-Play Design with No Test-Time Fine-tuning**: Offering versatility for various downstream applications and strong adaptability to real-world use cases (**affirmed by Reviewers DcWh, Q24f, MzvZ**).

**4. Clear Exposition and Rigorous Technical Design**: Clearly articulating the core challenges of PT2I and corresponding solutions, ensuring logical coherence and readability (**affirmed by Reviewers DcWh, hPcs, Q24f**).

***

## Core Concerns & Corresponding Revisions

We have systematically addressed all reviewer concerns through substantive revisions and detailed clarifications (all modifications are highlighted in **blue** in the revised manuscript):

**1. In-depth Analyses of Key Components**:

- Dynamic Decoupling Strategy (DDS) (**Reviewers DcWh, hPcs, Q24f, MzvZ**): Comprehensive analytical results are presented in **Sec.B.6** of the revised manuscript, which **have addressed the concerns of Reviewer Q24f and MzvZ**;

- Hierarchical Mixture-of-Experts Feature Fusion Module (HMoE-FFM) (**Reviewer hPcs**): Detailed technical analyses and corresponding results are provided in **Sec.B.7** of the revised manuscript;

- Decoupled Cross-Attention (DCA) (**Reviewer MzvZ in Round 2 review**): Detailed mechanistic analyses and illustrative examples have been supplemented in **Sec.B.7 (L1564-1646, page 29-31)** of the revised manuscript to resolve the remaining concern.

**2. Clarifications/Refinements to Evaluation Setups**:

- System prompt details for evaluation metrics (**Reviewer Q24f**): Complete system prompts for evaluation metrics are provided in **Figs. 7 and 8** of the revised manuscript;

- Prompt following metric does not consider all aspects of the given instructions (**Reviewer Q24f in Round 2 review**): This concern arose from a **misunderstanding**. We have provided precise and detailed clarifications to fully resolve it;

- Potential biases from single VLM evaluation (**Reviewer Q24f**): We have integrated **multiple SOTA VLMs** into our evaluation pipeline and **updated all relevant scores** in the revised manuscript, which **has addressed Reviewer Q24f’s concern**;

- Potential inconsistency in input prompt formats for instruction-based methods (**Reviewer MzvZ**): We have clarified that input consistency was strictly ensured by **adapting prompts to the officially recommended format** of each compared method, which **has addressed Reviewer MzvZ’s concern**;

- Zero-shot evaluation should train all compared methods on the same data (**Reviewer MzvZ in Round 2 review**): A detailed explanation of the experimental design rationale has been provided to address this concern.

**3. Supplementation of Technical Details**:

- Training and dataset details (**Reviewer DcWh**): Comprehensive details regarding model training, dataset curation, and preprocessing pipelines are provided in **Sec.B.1** of the revised manuscript;

- Additional ablation studies on alternative layer choices for HMoE-FFM (**Reviewer hPcs**): Extended ablation experiments are presented in **Sec.B.8** of the revised manuscript.

**4. Other Concerns:**

- Failure cases and limitations (**Reviewer DcWh**): A discussion on the limitations and failure cases of our approach is provided in **Sec. C** of the revised manuscript;

- Unsupported claims and wrong reference (**Reviewer hPcs**): We have revised the **relevant claims in L327-329** of the revised manuscript and **corrected** the erroneous reference;

- Relevant work similar to HMoE-FFM (**Reviewer MzvZ**): We have clarified the **fundamental differences** between HMoE-FFM and this work, which **has addressed Reviewer MzvZ’s concern**.

***

— To be continued in the next message.

---

> ### Author Response · Authors · 2025-12-03
> **Rebuttal Summary for Area Chairs**
>
> ## Summary
>
> Our work fills a critical gap in PT2I generation by tackling three core challenges (**CP·PF balance, fine-grained concept preservation, and multi-subject scalability**) through two novel and technically rigorous innovations: the Dynamic Decoupling Strategy (DDS) and HMoE-FFM. Following the rebuttal process, the manuscript has been substantially strengthened: we have **enhanced reproducibility by supplementing comprehensive technical details, deepened mechanistic clarity to resolve interpretability concerns, and improved evaluation rigor to align with community standards**.
>
> All reviewers have **affirmed the work’s core value, strong performance in comprehensive evaluations, practical impact, and clear exposition with rigorous design**. Aligned with ICLR’s focus on impactful and practical generative AI research, the revised manuscript meets the conference’s high standards for novelty, technical soundness, and methodological rigor. **We respectfully request your consideration for acceptance and stand ready to implement any further minor revisions to refine the work.** Thank you very much for your time and careful consideration!
>
> Sincerely,
>
> The Authors

---

### Meta-Review · Area_Chair_r2Jz · 2026-01-06

**Summary:**

This paper proposes DynaIP, a training-free personalized text-to-image (PT2I) adapter for Multimodal Diffusion Transformers (MM-DiTs). The method introduces a Dynamic Decoupling Strategy (DDS) to disentangle concept-specific information from concept-agnostic features  and a Hierarchical Mixture-of-Experts Feature Fusion Module (HMoE-FFM) that adaptively integrates multi-level CLIP features to enhance fine-grained visual details and support scalable multi-subject generation.

The main concerns with the reviewers are lack of analysis, missing details, evaluation metrics not fully convincing and novelty being limited. The authors have attempted to address many of these concerns in the rebuttal. But, I think the paper still misses rigorous analysis and understanding of the method. The novelty, in my opinion, is somewhat limited as well, which is fine. But, the experimental results should have been more detailed and comprehensive. I would strongly encourage the authors to work on this, and have a strong experimental section in the next version.

Because of these concerns, I am leaning towards rejecting this paper.

**Reviewer Concerns:**

Some of the main concerns from the reviewers are
[DcWh]
1. Policy model training details not fully mentioned. What is the exact reward function used? How sensitive is the method to the choice of policy architecture or optimization hyperparameters? How much training data is needed to train the policy effectively?
[hPcs]
2. Limited post-hoc analysis of key components
[Q24f]
2. Evaluation metrics is not very convincing. The high PF score of 0.997 stems from the fact that the emphasis was on subject consistency.
3. Missing comparisons on Qwen-Image and SD3
[MzvZ]
4. Prompts employed by FLUX.1 Kontext Dev and Qwen-Image-Edit adopt an instruction-based format, making the comparison in the paper potentially unfair.
5. Novelty is limited. Work is similar to "Multi-Layer Visual Feature Fusion in Multimodal LLMs: Methods, Analysis, and Best Practices]"
6. Method is not strictly zero-shot. The proposed method trains additional components such as "HMoE-FFM" and "Cross-attention", whereas the instruction-based models (FLUX.1 Kontext Dev, Qwen-Image-Edit, OmniGen2, ...) were not trained on the same data.

**Reviewer Scores:**

[DcWh]
In the rebuttal, the authors included some additional details about the training. But these are not as comprehensive as what DcWh asked. So, I think DcWh would have stayed on the same score.
[hPcs]
The authors have included some analysis and visualization in the rebuttal. But, they are not very comprehensive. For this paper to be a strong submission, a lot of analysis on disentanglement and understanding of why the method needs to be done, which is still missing. I think the reviewer hPcs would have stayed in the same score.
[Q24f]
Regarding the evaluation metric, the authors added an explanation of how they focus on text alignment in the system prompt of VLM. But, as the reviewer Q24f mentioned, all aspects of prompt alignment such as object, style, color, texture, etc are still missing. In the initial assesment, the reviewer maintained the score of 4 and I think they would have maintained the same score post rebuttal as well.
[MzvZ]
Regarding point 4, the reviewers clarified it clearly in the rebuttal.
For limited novelty, the authors clarified the differences from the prior work in the rebuttal.
Regarding method not being zero-shot, the authors clarified this point as well. I agree with the authors - the method trains on general datasets and tests on other eval datasets. I am not sure about the terminology of zero-shot, but at least it doesn't overfit on the eval set and the training-test difference is on par with methiods like Qwen-Image edit.
It is possible that reviewer MzvZ might have increased their score to 5 since some of their concerns were addressed.

---

### Decision · Program_Chairs · 2026-01-26

Reject